

# Analysis and evaluation of WRF microphysical schemes for deep moist convection over Southeastern South America (SESA) using microwave satellite observations and radiative transfer simulations

**Victoria Sol Galligani[1,2], Die Wang[3], Milagros Alvarez Imaz[1,2], Paola Salio[1,2,4], and Catherine Prigent[3]**

[1]Centro de Investigaciones del Mar y la Atmsfera, CONICET-UBA, Buenos Aires, Argentina.
[2]UMI-Instituto Franco Argentino sobre Estudios del Clima y sus Impactos, Buenos Aires, Argentina.
[3]Laboratoire d'Etudes du Rayonnement et de la Matire en Astrophysique (LERMA), CNRS, Observatoire de Paris, Paris, France.
[4]Departamento de Ciencias de la Atmsfera y los Ocanos. FCEN. Universidad de Buenos Aires. Buenos Aires, Argentina.

Corresponding author: Victoria Galligani, victoria.galliganil@cima.fcen.uba.ar





## Abstract

In the present study, three meteorological events of extreme deep moist convection, characteristic of South Easter South America, are considered to conduct a systematic evaluation of the microphysical parametrizations available in the Weather Research and Forecasting (WRF) model by undertaking a direct comparison between satellite-based simulated and observed microwave radiances. A research radiative transfer model, the Atmospheric Radiative Transfer Simulator (ARTS), is coupled with the Weather Research and Forecasting (WRF) model under three different microphysical paramterizations (WSM6, WDM6 and Thompson schemes). Microwave radiometry has shown a promising ability in the characterization of frozen hydrometeors. At high microwave frequencies, however, frozen hydrometeors significantly scatter radiation, and the relationship between radiation and hydrometeor populations becomes very complex. The main difficulty in microwave remote sensing of frozen hydrometeor characterization is correctly characterizing this scattering signal due to the complex and variable nature of the size, composition and shape of frozen hydrometeors. The present study further aims at improving the understanding of frozen hydrometeor optical properties characteristic of deep moist convection events in South Easter South America. In the present study, bulk optical properties are computed by integrating the single scattering properties of the Liu (2008) DDA single scattering database across the particle size distributions parametrized by the different WRF schemes in a consistent manner, introducing the equal-mass approach. The equal mass approach consists in describing the optical properties of the WRF snow and graupel hydrometeors with the optical properties of habits in the DDA database whose dimensions might be different ($D_{max}^{'}$) but whose mass is conserved. The performance of the radiative transfer simulations is evaluated by comparing the simulations with the available coincident microwave observations up to 190 GHz (with observations from TMI, MHS, and SSMI/S) using the Chi-square test. Good agreement is obtained with all observations provided special care is taken to represent the scattering properties of the snow and graupel species.

## 1 Introduction

The continental region east of the Andes, covering the south of Brazil, Paraguay, Uruguay, and the north and centre of Argentina (usually referred to as South Eastern South America, SESA), is known for its large and intense Mesoscale Convective Systems (MCSs) within which severe weather events develop (e.g., *Altinger de Schwarzkopf and Necco* [1988], *Silva Dias* [2011], *Mezher and Barros* [2012], *Goodman et al.* [2013], *Salio et al.* [2015]). These are the regions where the strongest MCSs on Earth occur [*Zipser et al.*, 2006]. In this data sparse region, little is known about the aspects of these systems, including what governs their structure, life cycle, similarities and differences with severe weather-producing systems observed elsewhere on the Earth, and their predictability from minutes to climate time-scales. High resolution models are a powerful tool to study convection.

NWP models can be used to perform numerical experiments in controlled environmental conditions, to assess the impact of different physical processes and environmental conditions upon the life cycle and the organization of convection (e.g., *Morrison and Khvorostyanov* [2005], among others). The description of cloud processes and ultimately the dynamical processes that result from numerical models need to be improved to more accurately describe key factors such as hydrometeor characteristics, latent heating profiles, radiative fluxes and forcing, entrainment, and cloud updraft and downdraft properties. This is particularly important since, with the increase of computing power in the recent years, the physical parameterizations in climate and numerical weather prediction (NWP) models have improved to incorporate microphysical processes, often at increasingly high resolution, resolving the dynamical interactions in convective systems.



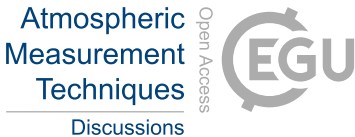

Cloud resolving models can be operated with different parameterizations, including different microphysics schemes. In recent years, increasingly detailed bulk cloud microphysics parameterizations have been incorporated into cloud resolving models. Bulk microphysics represent the size spectra of the different hydrometeor species with a particle size distribution function. In this way, microphysics parameterizations predict the development of one or more hydrometeor categories, their interactions and growth, and precipitation. Microphysics schemes may differ in the number of predicted species, predicted moments, number of simulated microphysical processes, assumptions regarding the mass-size relationships and size-terminal fall speed relationships, and the assumed particle size distributions. An extensive evaluation of the existing schemes is needed in order to constrain and reduce the uncertainties associated with the parameterizations. The microphysical properties (e.g., dielectric properties, density, particle size distribution, shape, orientation) of the frozen particles specifically, have a very complex temporal and spatial variability, and lack robust parameterizations.

Microwave radiometry has shown a promising ability in the characterization of frozen particles, as it is able to penetrate and provide insight into the vertical profiles of most clouds, in contrast to infrared and visible observations, which essentially sense cloud tops. At low microwave frequencies, hydrometeors essentially interact with the radiation through emission and absorption. These interactions are well parameterized using only simple assumptions. In contrast, at high microwave frequencies, frozen hydrometeors can significantly scatter radiation, and the relationship between radiation and hydrometeor populations becomes much more complex. In the model-to-satellite approach, satellite radiances are simulated using outputs from atmospheric models and compared to available observations using a radiative transfer model (e.g. *Chaboureau et al.* [2008]; *Meirold-Mautner et al.* [2007], *Galligani et al.* [2014]). Under cloudy conditions and at high microwave frequencies (> 80 GHz), the radiative transfer calculations are more difficult to handle and they strongly depend upon a much more detailed description of the cloud microphysics than the parameterizations that are currently available in NWP models.

In the present study, meteorological events of extreme deep moist convection are considered to conduct a systematic evaluation of the micro-physical parametrizations available in the Weather Research and Forecasting (WRF) model. In order to do this, a direct comparison between satellite-based simulated and observed microwave radiances is proposed by coupling the WRF model with a research radiative transfer model, the Atmospheric Radiative Transfer Simulator (ARTS). Since the simulation of passive microwave radiances requires good knowledge of the scattering properties of frozen hydrometeors, the present study further aims at improving the understanding of frozen hydrometeor optical properties and the characteristics of deep convection in the SESA region. This study is structured as follows. Section 2 introduces a particular deep moist convection event in the SESA region, together with a description of the models used and the available microwave observations. This section includes a discussion of the modelling system developed in the present study that converts WRF outputs to simulated microwave brightness temperatures (TBs). Section 3 focuses on the difficulties associated with providing the radiative transfer model used with a rather accurate description of the radiative properties of the hydrometeors modelled by WRF, especially for frozen hydrometeors. A sensitivity study of the passive radiative transfer simulations to the hydrometeor characteristics is presented in Section 4 for specific observed transects, followed by a statistical analysis of the simulated and observed brightness temperature distributions. Section 5 further tests the drawn conclusions by simulating two other convective events in the region. Finally, Section 6 presents the conclusions and details future work being carried out to exploit this modelling system.



## 2 A severe weather event associated with deep convection in the SESA region: models and observations

The focus of the present study is an intense MCS event observed over the centre of Argentina on 6 December 2012. On the 6 and 7 of December, the center of Argentina was affected by many severe weather events, including tornadoes, winds above 100 km/hr, and intense precipitation that caused tragic floods in the city of Buenos Aires. The following sub-sections describe the observations available during this meteorological event, the configuration used in the WRF model runs and its microphysics parameterizations, and the radiative transfer model used.

### 2.1 Coincident satellite observations

For the MCS event on the 6 December 2012 there are coincident observations available from the Tropical Rainfall Measuring Mission (TRMM) at 07:00 UTC and the Microwave Humidity Sounder (MHS) onboard NOAA-19 at 19:00 UTC. TRMM carries a suite of instruments designed to study precipitation in the tropics (*Kummerow et al.* [1998]). The TRMM Microwave Imager (TMI) is a conical imager operating at 10.7, 19.4, 21.3, 37, and 85.5 GHz with a $53^o$ incidence angle. It has two orthogonal polarizations (except at 22 GHz) and spatial resolutions between 63 km x 37 km, and 7km x 5km, depending on the channel. It covers a swath of 780 km. The TRMM Precipitation Radar (PR) operates at 13.8 GHz with a 4 km resolution and a swath of 220 km. The swath is located in the center of the TMI swath. The Microwave Humidity Sounder (MHS) is a cross-track sounder with surface zenith angles varying between $0^o$ and $58^o$. The channels are located at 89.0, 157.0, 183.3 ± 1, 183.3 ± 3 and 190.3 GHz. The channels near the water vapour line of 183.3 GHz are strongly sensitive to atmospheric absorption, in contrast to the more transparent window channels at 89, 157 and 190 GHz. The spatial resolution at nadir is 16 km for all channels and increases away from nadir (26 km at the furthest zenith angle along track). The polarization state for each channel is a combination between the two orthogonal linear polarizations (V and H), where the polarization mixing depends on the scanning angle. TMI observations at 10.7, 19.4, 37 and 85.5 GHz are shown in Figure 1(a-d) for vertical (V) polarizations only. The highly scattering MCS event is evidenced by brightness temperature depressions at the higher frequency channels ($> 37$ GHz). At the lower frequency channels ($< 37$ GHz), TMI is mostly sensitive to surface emission. The ocean surface emissivities are rather low and polarized, contrarily to land surfaces that usually have a high emissivity with limited polarization. For low atmospheric opacity at the lower frequencies, the contrast between ocean and land is larger. This contrast can easily be seen up to 19 GHz in Figure 1. At 37 GHz, both liquid water emission in clouds and frozen hydrometeor scattering induce a decrease in TB over the highly emitting land. At the higher frequency channel of 85.5 GHz, cloud structures appear cold due to the strong scattering of frozen hydrometeors, with rather low TBs (down to almost 50 K on this case study). Figure 1(e) shows the PR reflectivity and the PR retrieved freezing level height crossing the MCS system along the black line shown in Figure 1(d). MHS observations at 89, 157, 183 ± 1 and 190 GHz are shown in Figure 2(a-d) (the 183 ± 3 is very similar to the 190 GHz channel and is not shown). Note that MHS zenith angles vary between $58^o$ (on the west) and $0^o$ (on the east). In the window channels, the observations over the ocean present rather low brightness temperatures due to the low ocean emissivity when compared to those over the continental region. With increasing atmospheric opacity in the $H_2O$ water vapor line, as evidenced at 183 ± 1 GHz, the contrast between land and ocean disappears. In the window channels, the scattering effect due to the presence of convection can be observed from the brightness temperatures depressions that increase with frequency, especially in the window channels. The strong brightness temperature depressions that are even observed in the water vapour line channel (TBs≈100 K) evidence the presence of highly scattering clouds. The following subsections described the models exploited to use this meteorological event in a systematic evaluation of microphysical parameterizations.





## 2.2 The mesoscale cloud model: The Weather Research and Forecasting (WRF) model and the WSM6, WDM6 and THOM microphysics options

WRF is a non-hydrostatic mesoscale numerical weather prediction system designed for both atmospheric research and operational forecasting needs. It provides a full description of the atmospheric parameters (pressure, temperature, and mixing ratios for the water vapor, and the five hydrometeor categories). In the present study, WRF-ARW (*Skamarock and Klemp* [2008]) version 3.6 is used for the model simulations considering only one domain with 4 km grid spacing and 38 vertical levels. The model was initialized with GFS (Global Forecast System) initial conditions of $0.5^o$ resolution at 00:00 UTC for 6 December 2012. The model was integrated up to 36 hours with every 3 hour updates of the boundary conditions taken from GFS analysis also at $0.5^o$ resolution. Figure 3 shows the domain considered and Table 1 presents the different parametrizations used in the model run.

The three microphysics schemes used in the present study include the WRF Single-Moment 6 (WSM6; *Hong and Lim* [2006]), the WRF Double-Moment 6 (WDM6; *Hong et al.* [2010]) and the Thompson schemes (THOM, *Thompson et al.* [2008]). The three schemes have the same number of water species (water vapour, cloud water, rainwater, cloud ice, snow, and graupel). The WSM6 is a single-moment scheme that prognoses the mass mixing ratio of species, whereas the WDM6 is a double moment scheme based on the WSM6 that additionally prognoses the number concentration mixing ratios of cloud water and rainwater related to the size distribution of the species, i.e., double-moment representation of warm-rain. The THOM scheme also additionally prognoses number concentration mixing ratios for cloud ice and warm-rain.

These microphysics schemes generally assume a gamma particle size distribution (PSD) for precipitating hydrometeor species of the form:

$$N_x(D) = \int N_{0x} D^{\mu_x} e^{\lambda_x D} dD, \tag{1}$$

where $N_x(D)$ represents the number concentration ($\mathrm{m^{-1}m^{-3}}$) of particles of a given hydrometeor class (x) and diameter D, $N_{0x}$ is the y-intercept parameter, $\lambda_x$ is the slope parameter, and $\mu_x$ is the shape parameter of the distribution. This gamma distribution is simplified to an exponential distribution by setting $\mu_x$ to zero for rainwater, snow, and graupel in both the WSM6 and WDM6 schemes, and for rainwater and graupel in the THOM scheme. Snow is unique in the THOM scheme because, in contrast to most WRF bulk schemes, its particle size distribution is not an exponential size distribution, but a sum of two gamma functions following observations by *Field et al.* [2005]. The particle size distribution, hereafter referred to as the *Field et al.* [2005] size distribution, is based on in-situ observations valid for tropical and midlatitude clouds, and has been used with positive results in recent validation studies (e.g. *Doherty et al.* [2007]; *Kulie et al.* [2010]). Additionally, snow mass (and indirectly density) in the THOM scheme is not fixed and varies inversely with diameter D as m(D)=0.069D² unlike most schemes, including the WSM6 and WDM6 schemes, that have a fixed mass determined by m(D)=$(\rho_s\pi/6)$D³ where $\rho_s$=0.1kg/m³. This is an important difference since observational studies rarely support fixed density snow habits. *Magono* [1965] and many later studies recognize that a size-independent density is not a physically sound assumption for snowflakes because of the rigidity of ice and the nature of the snow formation processes (*Leinonen et al.* [2012]). The *Field et al.* [2005] PSD takes into account the parameters of the mass-size relationship and predicts a higher number of smaller particles, but a smaller number of larger particles than the WSM6/WDM6 schemes. It is also worth stating that the graupel species in the THOM scheme represent rimed ice (e.g., hybrid like graupel−hail category) by





using a two-parameter diagnostic dependence of its size distribution intercept param-
eter based on the mass mixing ratio and amount of supercooled liquid water.

Figure 4 shows the integrated column contents in kg/m$^2$ for rain (4a-c), snow (4d-
f) and graupel (4g-i), as simulated by the three different schemes at 19:00 UTC with a
minimum threshold of 0.05 kg/m$^2$. Note that the integrated contents for ice cloud and
cloud water are not shown. This specific time output corresponds to the over-pass of the
Microwave Humidity Sounder (MHS) discussed above. Another time output considered
in the present study is the TRMM overpass at 07:00 UTC (not shown). The black line
in Figure 4g represents an MHS transect simulated which is explored in Section 4. A first
look at Figure 4 shows that the three schemes model the structure and the location of
the cloud system fairly similarly. The brightness temperature depressions observed in
Figure 2 (and Figure 1) correspond to the cloud structures simulated by WRF in Fig-
ure 4 at 19:00 UTC (and at 07:00 UTC not shown). A close examination of MHS ob-
servations (Figure 2) and the WRF cloud outputs (Figure 4), however, reveals that the
cloud system modelled by WRF is slightly time lagged and misplaced with respect to
the observations, similarly to TMI observations (Figure 1) and the corresponding WRF
cloud outputs (not shown). A closer look at the mass loading of the different hydrom-
eteor also evidences a strong sensitivity to the microphysical scheme used. As expected,
the WSM6 and the WDM6 schemes model similar hydrometeor mass loadings. The THOM
scheme, on the other hand, shows much higher snow contents. Figure 5 further shows
the domain-averaged vertical distribution of the hydrometeor contents modelled by the
different schemes between 18:00 and 19:00 UTC. Units are in g/kg for all the species.
Both Figure 4 and Figure 5 show a comparable behaviour in the frozen phase (ice, snow
and graupel) in the WSM6 and WDM6 schemes. This is expected because the WDM6
scheme follows the cold-rain processes of the WSM6 scheme and the added processes in
the WDM6 do not affect the frozen phases directly (*Lim and Hong* [2010]). Figure 5 shows
an increase of the WDM6 rainwater mixing ratio below 5 km with less cloud droplet mix-
ing ratios. The THOM scheme, as previously reported by e.g., *Kim et al.* [2013], is dom-
inated by snow throughout the vertical profile and predicts the smallest amount of rain
water. The THOM scheme has a maximum cloud water content between 8 and 10 km.
This peak of enhanced cloud water content is found within and around strong convec-
tive updrafts. In order to compare the distribution of the frozen hydrometeor species among
the total frozen phase for each scheme, Figure 5 additionally shows the mean vertical
profile of the total frozen content (i.e., ice+snow+graupel, shown in light blue). The to-
tal frozen content is comparable in magnitude in all the schemes analyzed but since each
scheme has different intrinsic assumed characteristics and microphysical processes, they
partition the total content in different ways between graupel, cloud ice, and snow. The
THOM scheme has the most prominent vertical structure. Note that very similar remarks
can be drawn from the model simulations at 07:00 UTC in coincidence with the avail-
able TMI observations (not shown).

### 2.3 The radiative transfer model: The atmospheric radiative transfer simulator (ARTS)

A robust radiative transfer model allows consistently modelling passive observa-
tions when using (1) WRF outputs to describe the atmospheric profiles as discussed above
and, (2) a rather accurate description of the radiative properties of the hydrometeors in
each model grid point. In the present study, the Atmospheric Radiative Transfer Sim-
ulator (ARTS, *Eriksson et al.* [2011]) is used. ARTS is a very flexible tool, capable of
modeling different atmospheric conditions and different sensor configurations. ARTS is
an open-source code available at http://www.radiativetransfer.org along with extensive
documentation. It is a well validated model (*Melsheimer et al.* [2005], *Buehler et al.* [2006],
*Saunders et al.* [2007]) and it can handle scattering with arbitrary complex scattering
properties set by the users. It provides a Monte Carlo module to solve the radiative trans-



fer equation under cloudy conditions (*Davis et al.* [2007]) which takes full account of the 3-D description of the atmospheric state modelled by the WRF outputs.

To accurately simulate real microwave observations of satellite-based instruments with ARTS a correct description of the surface properties, the observation geometry and the cloud optical properties is important. The proposed methodology involves a series of coupling tools. The Tool to Estimate Land Surface Emissivities from Microwave to Sub-millimeter waves (TELSEM2; *Wang et al.* [2016]) and the Tool to Estimate Sea Surface Emissivity from Microwave to Sub-millimeter waves (TESSEM2; *Prigent et al.* [2016]) are used to determine land and ocean surface emissivities respectively. TELSEM2 provides the emissivity (V and H components) for any location, any month, and any incidence angle with a spatial resolution of 0.25 degrees. TESSEM calculates sea surface emissivities from wind, sea surface temperature and viewing angle. Coupling WRF outputs with ARTS further requires a good description of the hydrometeor optical properties (i.e., the single scattering properties) and particle size distributions. Bulk optical properties are computed by integrating the single scattering properties of particles across a given particle size distribution. The bulk optical properties of the hydrometeors at each model level have a strong influence on the radiative transfer equation for both passive and active simulations. The single scattering properties are determined by hydrometeor composition, density, dielectric properties, size, shape and orientation. While the particle size distribution of species is intrinsic to each WRF microphysics scheme, cloud resolving models like WRF do not determine all of the parameters needed to determine the single scattering properties, and further assumptions are necessary. This is discussed in more detail in Section 3 below.

## 3 Modelling the single scattering properties

Throughout the present study, the goal when implementing the single scattering properties and the particle size distribution of the hydrometeor species in ARTS is to remain as consistent as possible with the corresponding WRF microphysics scheme. The particle size distributions for each hydrometeor category in the radiative transfer simulations remains consistent with the parameterizations used in the WRF. The single scattering properties of hydrometeors, on the other hand, require assumptions to be made.

For simplicity, the optical properties of cloud ice, cloud water and rain are held constant and represented by Mie spheres with the dielectric properties of *Liebe et al.* [1991] for liquid species and *Mätzler* [2006] for ice crystals. These are reasonable assumptions for the liquid phase. The mass loadings of ice crystals simulated by WRF in the scenes explored are negligible and, at the microwave frequencies analysed, small pure ice crystals produce very little scattering. Modelling snow and graupel species, on the other hand, in much more challenging, mainly due to uncertainties in their composition and shape. Frozen hydrometeors have a large spatial and temporal variability and are of a complex non-spherical nature. Frozen hydrometeors can be both single crystals (with shapes including needles, plates, columns, rosettes, dendrites, etc.) or aggregates (e.g., *Baran* [2012]). There is a highly complex mixture of differently shaped and sized habits in the atmosphere, and this mixture further varies with particle size. However, the only computationally realistic approach is to assume a one-shape model to represent the total habit population even if this approach does not fully capture the large variability observed in nature.

There are a number of approaches used to model frozen hydrometeors. One is to assume that the habits have certain known realistic shapes like plates or rosettes, and calculate their single scattering properties using the Discrete Dipole Approximation method (DDA, *Draine and Flatau* [1994]). The second approach is to approximate these complex shapes with spheres with the same mass and apply Mie theory. This imaginary sphere can either be a pure ice sphere with a smaller diameter or a "soft sphere" of the same




size but with lower density and a reduced effective dielectric constant. In the soft sphere
approximation, particles are considered to be homogeneous mixtures of ice/air, or pos-
sibly ice/air/liquid water. This approach requires that the mass fraction of, for exam-
ple air in the ice/air mixture and the corresponding dielectric properties of the homo-
geneous mixture, be determined. The soft sphere approximation has been widely used
together with the T-matrix method to model spheres and spheroids (e.g., *Galligani et al.*
[2014]), where the air fraction was either set to be fixed or derived from mass-size parametriza-
tions or snow habit densities. This approach, however, has been shown to be problem-
atic, as the air fraction in the mixing rule must be allowed to vary with both particle size
and frequency for a better fit (e.g., *Galligani et al.* [2014], *Eriksson et al.* [2015]). *Liu*
[2004] showed that the optimal softness parameter, or effective density, varies with fre-
quency. However, using density-based air fractions which are a function of frequency and
size is an unphysical approach. Furthermore, for large particles in the more realistic size
dependent mass parametrizations as in the THOM scheme, it has been observed that
the larger particles have high air fractions and consequently negligible scattering efficien-
cies (e.g., *Galligani et al.* [2014]). Although the DDA approach can accurately evaluate
the radiative properties of more realistic, complex shapes, choosing a particular shape
model remains arbitrary and hence problematic. Readers are encouraged to refer to *Eriks-*
*son et al.* [2015] for a detailed discussion on the microwave optical properties of ice hy-
drometeors.

In this study, snow and graupel hydrometeors are modelled using scattering prop-
erties of realistic snowflake habits from the *Liu* [2008] database. *Liu* [2008] used the DDA
code of *Draine and Flatau* [1994] to compute the single scattering properties of differ-
ently shaped ice crystals. The *Liu* [2008] database presents 11 different randomly ori-
ented ice crystals at 22 frequencies (3.0 - 340 GHz) and at 5 different temperatures (233,
243, 253, 263 and 273 K). The main properties of the database are listed in Table 2. The
soft sphere approximation is also used for comparison, and following the conclusions drawn
in *Eriksson et al.* [2015], the *Maxwell-Garnett* [1906] mixing rule for air in ice is used to
model the effective dielectric properties, as it appears to have the least deviation from
DDA scattering properties.

The snow and graupel contents are thus described by the corresponding WRF par-
ticle size distribution and their single scattering properties by the *Liu* [2008] database.
One last remark must be made when using the *Liu* [2008] database to describe the scat-
tering properties of snow and graupel consistently with the WRF microphysics param-
terizations. Both the DDA habits and the WRF schemes use a mass-size relationship of
the form

$$m = aD_{max}^b, \qquad (2)$$

where a and b are parameters intrinsic to each of the DDA habits in the *Liu* [2008] database
or each of the hydrometeor species in the microphysics schemes, and indirectly deter-
mine the habit density. As described in Section 2.2, the snow mass in the THOM scheme
is not fixed with size and follows m(D)=0.069D$^2$ while the WSM6 and WDM6 schemes
have a constant mass value determined by m(D)=($\rho_s\pi/6$)D$^3$ where $\rho_s$=0.1kg/m$^3$. Grau-
pel species in the WSM6, WDM6 and THOM schemes have a constant density of $\rho_g$=0.4kg/m$^3$
and follow m(D)=($\rho_g\pi/6$)D$^3$. Similarly, each of the *Liu* [2008] habits are described by
different a and b parameters listed in Table 2. In order to consistently simulate WRF
model outputs with the *Liu* [2008] habits, the approach used in the present study is to
assume an equal mass habit where

$$a_{WRF}D_{max}^{b_{WRF}} = a_{LIU}D_{max}^{'b_{LIU}}. \qquad (3)$$

In Equation 3, $D_{max}$ is inferred from WRF parametrizations and is used in the parti-
cle size distribution. $D'_{max}$ is the corresponding equal mass DDA habit size used to de-
scribe the scattering properties of the WRF species consistently. This discussion is im-
portant since particle size is a key parameter in single scattering calculations. Figures



6(a) and (b) shows the corresponding equal mass $D'_{max}$ for a selected number of $Liu$ [2008]
habits when using the WSM6/WDM6 and THOM schemes respectively. The choice of
DDA habits shown is a result of regrouping certain habits that behave similarly, such
as the Thin hexagonal column, the Long hexagonal column, the Short hexagonal column
and the Thick hexagonal column, or the bullet rosettes. Note that the included black
dashed line represents unity. As shown in Figures 6(a) and 6(b), for a given maximum
particle dimension in WRF, the equal mass DDA habit $D'_{max}$ can be very different for
each of the $Liu$ [2008] habits. Figures 6(a) and 6(b) also show that equal mass DDA habit
$D'_{max}$ is larger when using the WSM6 and WDM6 schemes than when using the THOM
scheme. This is expected due to the intrinsic $\rho_s$ differences in these schemes. For the most
compact habits of the DDA database, like columns and plates, the difference between
the WSM6/WDM6 and the THOM schemes is the smallest, while the largest differences
are seen for the dendrite and sector habits. The thin hexagonal plates for example, have
$D'_{max}$ diameters above $D_{max}$ for the WSM6/WDM6, and $D'_{max}$ diameters below $D_{max}$
in the THOM scheme. The 6-b rosette $D'_{max}$ is larger for the WSM6/WDM6 schemes
but close to unity for the THOM scheme.

The bulk scattering properties (e.g., the extinction coefficient $\beta_e$) of each of the $Liu$
[2008] habits are shown in Figures 6(c) as a function of snow water content at 150 GHz
for the WSM6/WDM6 and the $Field\ et\ al.$ [2005] snow particle size distributions. This
$\beta_e$ parameter is calculated by integrating the extinction cross section $\sigma_e(D)$ across the
particle size distribution N(D):

$$\beta_e = \int_0^\infty \sigma_e(D)N(D)dD. \qquad (4)$$

As expected, extinction (and scattering) increases with frequency (not shown) and snow
water content. Not shown is the asymmetry parameter which gives an overall descrip-
tion of the phase function, i.e., the angular redistribution of scattered radiation. In con-
trast to the $Liu$ [2008] habits, the low density Mie sphere model (not shown) gives very
strong forward scattering for high snow water contents. The $Liu$ [2008] habits produce
more balanced forward and backward scattering. Although not shown graphically, analysing
the sensitivity of these bulk scattering properties with frequency indicates that these con-
clusions are broadly true for the microwave range of interest in the present study. As the
scattering increases, so do the differences between the bulk WSM6/WDM6 and THOM
properties. The integrated bulk properties showed in Figure 6(c) include the effects of
using the equal mass habit approach discussed above. Both the particle size distribu-
tions and how $D'_{max}$ differs from $D_{max}$ play an important role. Figure 6(c) illustrates
the complex nature of evaluating the relative importance of these two effects. In the WSM6/WDM6
schemes, the thin hexagonal plates and the 6-b rosette are the most scattering habits,
while the 6-b rosette and the dendrite habits are the least scattering habits. The bulk
scattering properties using the WSM6 and WDM6 schemes lead to higher scattering than
when using the THOM scheme, specially for the the most compact particles like columns
and plates, which are the most scattering. The opposite is true with the less compact
dendrite habits.

## 4   Comparison of the simulations with coincident observations

The objective of the following radiative transfer simulations is to consistently sim-
ulate the brightness temperature depressions observed related to the frozen phase us-
ing WRF microphysical properties and the necessary additional assumptions, with the
aim of evaluating the different DDA habits and the WRF microphysics options for the
meteorological event described in Section 2. It is not to simulate the detailed spatial struc-
ture of the observations because, as seen by comparing Figure 2 and Figure 4, there are
differences in the location of the observed and modelled cloud system. This section pro-
poses to undertake a sensibility analysis of the compatibility of WRF outputs and its
intrinsic microphysics parametrizations with the $Liu$ [2008] DDA habits. The present





study does not aim to search for the 'best' Liu habit. As discussed in the previous sec-
tions, the radiative transfer simulations to be discussed depend mainly on (1) the inte-
grated species content modelled by WRF, (2) the microphysics parametrized in each WRF
scheme, and (3) the additional single scattering properties of the frozen phase, more specif-
ically of snow species and graupel as discussed in Section 3. The particle size distribu-
tion remains consistent to the WRF microphysics scheme of interest, unless specified oth-
erwise.

To focus on cloudy simulations, one must first achieve robust clear sky simulations.
For a quantitative comparison, the statistical distribution of the simulated and observed
brightness temperatures is evaluated for some selected channels of TMI (10V, 19V, 22V,
37V, 85V GHz) and MHS (157, 89 and 183 ± 1 GHz). The statistical distributions (not
shown) show a good agreement with the observed brightness temperatures under clear
sky conditions, confirming the reliability of the radiative transfer simulation inputs (e.g.,
the surface emissivity estimates used or the state of the atmosphere simulated by WRF).
For the highly surface sensitive 19 GHz channel and the water vapour sensitive 22 GHz
channel, good agreement is found for the WSM6, WDM6, and THOM schemes, with bi-
ases (observed-simulated) of approximately -3.55 K and 0.6 K respectively over land. For
the water vapour channel in MHS at 183 ± 1 GHz, the schemes used show biases between
436 -1.33 and -1.68 K over land. The analysis of the distributions of simulated and observed
brightness temperatures under both clear and cloudy conditions, specially in window chan-
nels, essentially shows that the largest differences between the observed and simulated
brightness temperatures and especially at higher frequencies, is located in the lower end
of the brightness temperature histograms where scattering is important. Characteris-
ing the scattering signal responsible for the largest brightness temperature differences
is the focus of the present study.

Figure 7 shows the WSM6, WDM6 and THOM simulated brightness temperatures
at 19V, 37V and 85V GHz for a specific TMI scan of the observations presented in Fig-
ure 1 for the 6th December at 7 UTC, at the initiation stages of the system. Figure 7
shows the brightness temperature simulations for the selected *Liu* [2008] DDA habits in
Figure 6. The bottom row of Figure 7 shows the corresponding integrated snow, grau-
pel and rain contents simulated by the different WRF schemes. The out-most right col-
umn shows the corresponding TMI observations and serves as a reference to analyse the
simulations.

As discussed above, the clear sky observations are well simulated by all schemes.
However, the simulation of brightness temperatures in the presence of high snow and/or
graupel contents is shown to be problematic. This is clearly evidenced in Figure 7 by the
large spread in the simulated brightness temperatures throughout the different schemes
and the different DDA habits used. As expected, Figure 7 shows that the higher the fre-
quency, the larger the brightness depression simulated and the larger the sensitivity to
the different DDA habits. The large sensitivity of the simulated TBs to the DDA habits
shown in Figure 7, illustrates how problematic the representation of snow/graupel scat-
tering can be. Excessive scattering means that WRF generates more snow than is ob-
served, that the radiative transfer model (and its necessary assumptions) simulates ex-
cessive scattering, or both.

At 10 GHz (not shown), there is little sensitivity to scattering, and the most promi-
nent feature is a strong brightness temperature drop at approximately -32.9° due to a
lake in central Uruguay. This is observed more prominently in simulations and not in
observations due to the simplified antenna pattern used in the simulations. Due to the
lack of sensitivity to scattering, there is little sensitivity to the different DDA habits or
WRF microphysics schemes at 10 GHz.

At 19 GHz, all DDA habits produce excessive scattering for the WSM6 and WDM6
simulations, where the dendrite and sector habits simulate the warmest TBs closest to



the observed reference TBs, and the thick hexagonal plates and the block, long and short hexagonal columns (not shown) are the most scattering habits, producing the coldest TBs, followed by the thin hexagonal plate and the rosettes (only the 6-b rosette is shown). On the other hand, all DDA habits in the THOM scheme simulations produce similar TB depressions to those observed. The large depression observed at 19 GHz in the WSM6/WDM6 simulations is due to the high IWP graupel contents simulated by WRF. Note that due to the small brightness temperature depressions simulated using the THOM scheme, the signal coming from the lake at approximately $-32.9^o$ can be observed at 19 GHz, while simulations using the WSM6/WDM6 schemes are dominated by excessive scattering and consequently cloud signals dominate all surface signals. Note that although the THOM scheme is predicting the largest amount of integrated snow content, it does not necessarily produce the largest brightness temperature depressions.

Similar conclusions can be drawn for the 37 GHz simulations. At 37 GHz, however, as expected, the sensibility to scattering increases and consequently TB depressions also increase. All habits, except the sector and dendrite habits, produced excessive scattering with the WSM6 and the WDM6 schemes. Under the WDM6 scheme simulations, DDA habits show a warmer TBs compared to the WSM6 scheme. This is due to the strong graupel contents simulated by the WSM6 scheme. In the WSM6 and WDM6 schemes, sector and dendrite habits simulate comparable TBs to those observed, while the thick hexagonal plates and the block, long and short hexagonal columns (not shown) are the most scattering habits, producing larger TB depressions to those observed, followed by the thin hexagonal plate and the rosettes (only the 6-b rosette is shown). For the THOM scheme simulations, the DDA habits show a smaller spread in simulated TBs, and these TBs are all comparable to the reference observations, except for the 3-b, 4-b and 5-b rosettes (not shown) and the sector habit.

Simulations at 85 GHz, as expected, show an even higher sensitivity to scattering. In general, the combination of WSM6 and WDM6 and the DDA habits analysed follow the same sensitivities because they have the same particle size distribution, the same snow and graupel density (0.1 and 0.4 kg/m$^3$), and similar snow and graupel column contents. In these schemes, and for all the frequencies analysed, the sector and dendrite habits scatter the least and produce TB depressions closest to the reference TMI observations. The thick hexagonal plates and the long, short and block hexagonal columns (not shown) scatter the most, followed by the thin hexagonal plate and rosettes (only the 6-b rosette is shown). These produce excessive scattering in comparison to the reference observations. As discussed in Section 3, the bulk DDA(THOM) scattering properties is different to the bulk DDA(WSM6/WDM6) scattering properties due to the different particle size distributions and mass-size relationships (see discussion in Section 3). This is illustrated in Figure 7 for the 85 GHz channel simulations. For the THOM scheme simulations, contrary to the WSM6 and WDM6 simulations, the thin hexagonal plate is simulating the warmer TBs (smallest TB depressions), while the sector habits are producing the coldest temperatures (largest TB depressions).

MHS simulations at higher frequencies provide higher sensitivity to the scattering properties. Figure 8, similarly to Figure 7, focuses on a specific MHS scan from close to nadir to its outermost angle east, characterized by a large snow content in the WRF simulations (see black line Figure 4g). This transect belongs to observations on the 6th December at 17 UTC shown in Figure 2, where the system is in its developed stage. Figure 8 shows the simulated brightness temperatures of MHS channels with the exception of the 183±3 GHz, as it is very similar to the 183±1 GHz due to its water vapour sensitivity, for the WSM6, WDM6 and THOM schemes. The bottom row of Figure 8 shows the corresponding integrated snow, graupel and rain contents simulated by the different WRF schemes and the outmost right column shows the corresponding reference MHS observations. MHS observations must be used as a reference and not as a direct com-



parison to the simulations due to differences in timing and spatial structure of the meteorological fields modelled by WRF.

As expected the higher the window channel, the largest the brightness temperature depressions. As analysed for the TMI transect, Figure 8 shows that for WSM6 and WDM6 simulations, the dendrite and sector habits are the least scattering habits, and for simulations with the THOM scheme, the dendrite and the thin hexagonal plates (and the thick hexagonal plates and the long, short and block hexagonal columns not shown), are the least scattering habits. The habits producing the largest brightness temperature depressions in the WSM6 and WDM6 schemes are the thick hexagonal plates and the long, short and block hexagonal columns (not shown), followed by the thin hexagonal plate and the rosettes (only the 6-b rosette is shown), as discussed for the TMI channels simulated. In the THOM scheme, the coldest TBs are observed for the sector habits and the thin hexagonal plates as shown in Figure 8, and the thick hexagonal plates and the long, short and block hexagonal columns (not shown), also as discussed for the TMI channels.

As shown in Figure 7 for the TMI simulations, the THOM scheme MHS simulations in Figure 8 show that, in contrast to the WSM6 and WDM6 scheme simulations, the thin hexagonal plate is producing the smallest brightness temperature depressions and the sector habit is producing the largest brightness temperature depressions. This is a result of the equal mass approach and the schemes particle size distributions.

Note that simulations using the soft sphere approximation and with a Mie theory with the corresponding WRF microphysics parameterized densities are included in Figure 8 (black dashed lines). The behaviour of the Mie sphere simulations compared with those of the DDA habits are very different with frequency, and are not scattering enough at large frequencies. Following *Liu* [2004], Mie theory can be used to reproduce the ensemble of the DDA database by adjusting the air fraction with frequency. This approach hence has no physical basis. It can be argued, however, that choosing one of the Liu habits to represent the highly complex and variable habit population is also problematic.

Figures 9 and 10 show a quantitative comparison of the simulated and observed brightness temperature distributions for the whole meteorological scene simulated for relevant TMI and MHS observations respectively. The statistical distributions of the brightness temperatures are shown for the observations (black line) and radiative transfer simulations of a selected group of DDA habits (colored lines consistent with Figures 6, 7 and 8). Note that only data over land, i.e., excluding coastal data and data over the ocean, is accounted in these distributions which are built with 5 K bins and where bins with less than 5 counts are neglected.

As expected, Figures 9 and 10 show that most departures between observations and simulations are associated with cloudy situations at low brightness temperatures. Figure 9 shows that, as expected, simulations at 10 GHz show little sensitivity to scattering. At the higher 19 GHz channel, the simulations start to show a larger sensitivity to the DDA habits. The simulations using the WDM6 scheme lead to excessive scattering at 19 and 37 GHz for all the habits shown. For the simulations with the WSM6 scheme, the thin hexagonal plate and the 6-b rosette show excessive scattering in comparison to observations at 37 and 89 GHz, while the sector and dendrite habits show a comparable distribution with those observed. Finally, simulations with the THOM scheme show comparable distributions to those observed for all DDA habits up to 37 GHz, while at 89 GHz the thin hexagonal plate and the dendrite habits behave similarly to the observations. Figure 10 shows further information to analyse the sensitivity to the choice of DDA habits using the higher frequency channels onboard MHS. Similarly to Figure 9, simulations with the WDM6/WSM6 scheme, and the thin hexagonal plate or the 6-b rosette show excessive scattering specially for the 89 and 157 GHz MHS frequency channels, while the sector habit produces a TB distribution closest to the observed distribution. Finally,





the simulations with the THOM scheme show that the sector and 6-b rosette produce
excessive scattering, while the dendrite and thin hexagonal plate produce distributions
closest to those observed. In general for the scene analysed, the dendrite habit performs
best for all the schemes. Similar results were obtained by *Geer and Baordo* [2014] when
analyzing the DDA shapes over land.

With the aim of analysing quantitatively the behaviour of the different DDA habits
under the three different microphysics schemes, the chi-square test is used. The chi-square
test is a verification method to evaluate how close the simulated distributions are to the
observed distributions. Figure 11 and 12 show the relative residuals $E_i$ computed for each
bin following:

$$E_i = [X(i) - Y(i)]/\sqrt{X(i)}, \tag{5}$$

where X(i) and Y(i) are the relative frequencies of observations and simulations respec-
tively for the ith bin of the TMI and MHS observations respectively. The histograms and
the $\chi^2 = \sum E_i^2$ values shown only take into account bins below 270 K (250 K for the
183±1 GHz) in order to neglect clear sky pixels and focus on the cloudy contribution.
Figure 11 and 12 further aid the analysis of Figure 9 and 10, to point at the performance
of the simulations using the different DDA habits with the different microphysics schemes.
The dendrite habits show low $\chi^2$ value across the microphysics schemes. In the WSM6
and WDM6 schemes, the sector snowflakes also perform well. The sector snowflakes, how-
ever, show very high $\chi^2$ values in the THOM scheme simulations. In the THOM scheme
simulations, the thin hexagonal plates follow the dendrite habits in the low $\chi^2$ values.

Finally, Figure 13 (Figure 14) shows TMI (MHS) observations in the first column,
followed by the radiative transfer simulations using the dendrite habits to describe the
scattering properties in the WSM6, WDM6 and THOM schemes (second, third and fourth
columns respectively). Despite errors in the location and coverage of the spatial struc-
tures of the cloudy fields modelled by WRF, the results depicted in Figure 13 shows that
radiative transfer simulations using the WSM6 and the THOM microphysics schemes
can be used to simulate the observed brightness temperature depressions provided spe-
cial care is taken to represent the scattering properties of the snow and graupel species.
At low microwave frequencies, Figure 13 shows that the WDM6 scheme leads to exces-
sive scattering at >19 GHz. Figure 14 shows good agreement between the three micro-
physics schemes and MHS observations.

## 5 Extending the radiative transfer simulations to two additional MCS events of interest

Two additional convective events in South Eastern South America are analysed in
this section in order to further test the validity of the above drawn conclusions. The two
events are observed over central Argentina on the 13 January 2011 and the 23 January
2014, and microwave observations are available from SSMI/S at 2200 UTC and MHS at
0200 UTC respectively. These observations are shown for the most scattering sensitive
channels in the first and second rows of Figure 15 for SSMI/S and MHS for a relevant
selection of instrument channels.

Figure 16 shows the integrated column contents in kg/m$^2$ with a minimum thresh-
old of 0.05kg/m$^2$, simulated by WRF for these two scenes at the time of the available
coincident observations. Figure 16 shows the strong sensitivity of the hydrometeor con-
tents to the WRF microphysical parametrizations, as discussed in Section 2.2. Similarly
to the WRF simulations analyzed in Section 2.2, Figure 16 shows that the WSM6 and
the WDM6 schemes model similar hydrometeor mass loadings for the iced species (i.e.,
snow, graupel and ice, not all shown), while the THOM scheme shows much higher snow
contents. Similarly to the scene analysed in the previous section, the WSM6 simulates
the largest amount of graupel content (not shown) followed by the WDM6 scheme. The
THOM scheme produces very little graupel contents. Note that the two scenes analysed





in this section are comparable in IWPs with the case analysed in Section 4. Similarly
to Section 3, it can also be said from Figures 15 and 16 that the microphysics schemes
in WRF model the structure and location of the cloudy system fairly well for these two
scenes too.

Radiative transfer simulations are performed for these two scenes in the same man-
ner as described in Section 3 and the histograms of the simulated and observed bright-
ness temperatures for the two scenes (not shown) are analysed. Analysing the scene on
the 13 January 2011 which has coincident SSMI/S observations, it can be shown that
at 19 GHz the radiative transfer simulations using all the DDA habits with the WSM6,
WDM6 and THOM schemes, result in similar TBs. Unlike the scene analysed in Sec-
tion 4, the WDM6 scheme in this scene does not show excessive scattering at 19 GHz.
At 37 GHz, however, the WDM6 simulations show a pronounced large population of sim-
ulations with brightness temperatures between 250 to 270 K for all habits. At 37 GHz,
the WDM6 scheme simulations show that the thin hexagonal plates and the 6-b rosettes
have the coldest brightness temperatures (largest TB depressions). These TB depres-
sions are unrealistically large compared to the coincident observations. In the WSM6 sim-
ulations, similarly to section 4, the thin hexagonal plate and the 6-b rosette habits are
responsible for the coldest brightness temperatures, while the dendrite and sector snowflakes
have warmer TBs and are closer to the observed brightness temperatures. The simulated
THOM scheme brightness temperatures, on the other hand, show that all DDA habit
simulations produce TBs that are very close to the observed TB distributions, as dis-
cussed for the simulations in Section 4.

For frequencies above 37 GHz, i.e., 91V, 150H and 183±6H GHz, since there is a
larger sensitivity to scattering, there is a larger sensitivity to the different habits. To aid
this discussion, the relative residuals $E_i$ are computed for this histograms in the same
way as described in Section 4, and their $\chi^2$ values shown in Figure 17(a). As shown in
Section 4, the THOM scheme simulations with the thin hexagonal plate and the den-
drite habits show the smallest $\chi^2$ values, while in the WSM6/WDM6 the dendrite and
sector snowflakes show the smallest $\chi^2$ values. Similar conclusions are drawn for the scene
with available coincident MHS observations, where the corresponding residuals and $\chi^2$
values calculated from the histograms of the brightness temperatures are shown in Fig-
ure 17(b). Note that only the most sensitive channels to scattering are shown, i.e., 89
GHz, 157 GHz and 190 GHz.

Finally, Figure 19 and 20 show that, as discussed for the MCS event simulated and
analysed in Section 4, radiative transfer simulations using the WSM6 and the THOM
microphysics schemes can be used to simulate the observed brightness temperature de-
pressions using the dendrite DDA habits to represent the scattering properties of the snow
and graupel species. In this scene, as discussed above, the WDM6 scheme is not observed
to produce excessive scattering at low microwave frequencies, but is shown to produce
warmer brightness temperatures than observed at MHS channels.

## 6 Conclusion

Three meteorological events of extreme deep moist convection, characteristic of South
Easter South America, have been considered in the present study to conduct a direct com-
parison between satellite-based simulated and observed microwave radiances, and to eval-
uate three different WRF microphysical schemes. In order to do this, a research radia-
tive transfer model, ARTS, has been coupled with the WRF model under the WSM6,
WDM6 and THOM microphysical parametrizations. Since the simulation of passive mi-
crowave radiances requires good knowledge of the scattering properties of frozen hydrom-
eteors, the present study has further aimed at improving the understanding of frozen hy-
drometeor optical properties and the characteristics of deep convection in the SESA re-
gion. Bulk optical properties are computed by integrating the single scattering proper-



ties of particles across a given particle size distribution. While the particle size distribution of species is intrinsic to each WRF microphysics scheme, cloud resolving models like WRF do not determine all of the parameters needed to determine the single scattering properties, and further assumptions are necessary. In this study the Liu (2008) DDA single scattering database, with 11 different iced habits, has been used to provide realistic scattering properties for snow and graupel species. In order to apply the optical properties of the Liu (2008) DDA database to the hydrometeor species modelled by the WRF microphysics schemes in a consistent manner, the equal-mass approach is introduced. The equal mass approach consists in describing the optical properties of the WRF snow and graupel hydrometeors with the optical properties of habits in the DDA database whose dimensions might be different ($D'_{max}$) but whose mass is conserved. The performance of the radiative transfer simulations have been evaluated by comparing the simulations with the available coincident microwave observations up to 190 GHz (with TMI, MHS, and SSMI/S). The systematic evaluation of WRF+ARTS radiative transfer simulations presents a tool to evaluate the representativity of the different WRF microphysics schemes.

In the present study, a strong sensitivity of the hydrometeor column contents to the choice of WRF microphysics scheme has been shown. The WSM6 and the WDM6 schemes model similar hydrometeor mass loadings for all iced species, while the THOM scheme shows higher snow contents. The WSM6 has been shown to simulate the largest amount of graupel contents followed by the WDM6 scheme, and finally the THOM scheme that produces very little graupel contents. An analysis of the domain-averaged vertical distribution of the hydrometeor contents, nonetheless, shows a comparable behaviour of the total ice phase (ice+snow+graupel) for the schemes analysed.

A direct comparison of the simulated and observed brightness temperatures shows that the microphysics schemes in WRF model the overall structure and location of the cloud system fairly well. The large sensibility to DDA habit choice shown in the simulated brightness temperatures, evidences the complexity in characterizing the frozen hydrometeors scattering signal and the importance of improving our knowledge in the subject. Although the present study has not aimed to search for the 'best' Liu habit, the statistical performance of the simulated brightness temperatures of the different Liu (2008) habits has been evaluated by analysing the histograms of the observed and simulated brightness temperatures, and using the chi-square test to evaluate how close the simulated distributions are to the observed distributions and hence the representativity of the different WRF microphysics schemes. The bulk scattering properties of the Liu (2008) habits are similar for the WSM6 and WDM6 schemes, but different to the THOM scheme. This is due to the different particle size distributions and mass-size relationships. This is reflected in the statistical analysis of the observed and simulated brightness temperatures. For example, the thin hexagonal plates are shown to be one of the least scattering habits in the THOM scheme simulations, but one of the most scattering in the WSM6/WDM6 simulations. The opposite is shown for the sector habits. Nonetheless, disregarding the observed detailed spatial structures, an overall agreement is obtained between the simulated and the observed brightness temperatures, provided that special attention is taken when describing the optical properties of snow and graupel species. The dendrite and the thin hexagonal plate habits show the smallest $\chi^2$ values for the THOM scheme WRF simulations, while the sector and dendrite habits show the the smallest $\chi^2$ values for the WSM6 and WDM6 schemes.

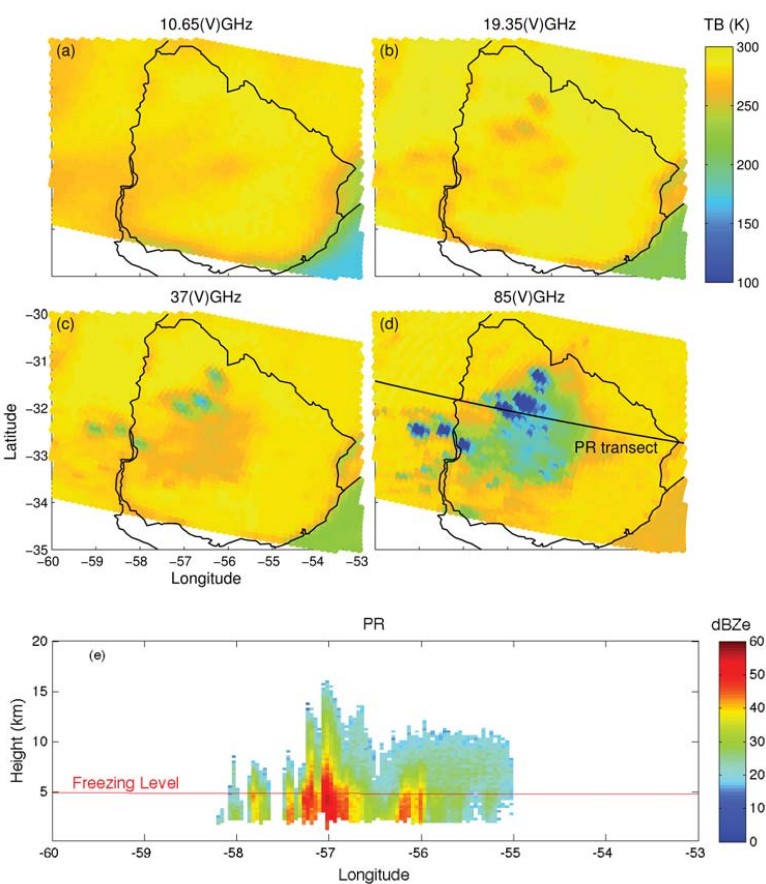

**Figure 1.** TMI observations on 6 December 2012 at 07:00 UTC for AN MCS event of interest in the present study. Note that the horizontally polarized channels and the 22V GHz channel observations are not shown. The solid black line in 1(d) represents the location of the PR transect shown in 1(e).



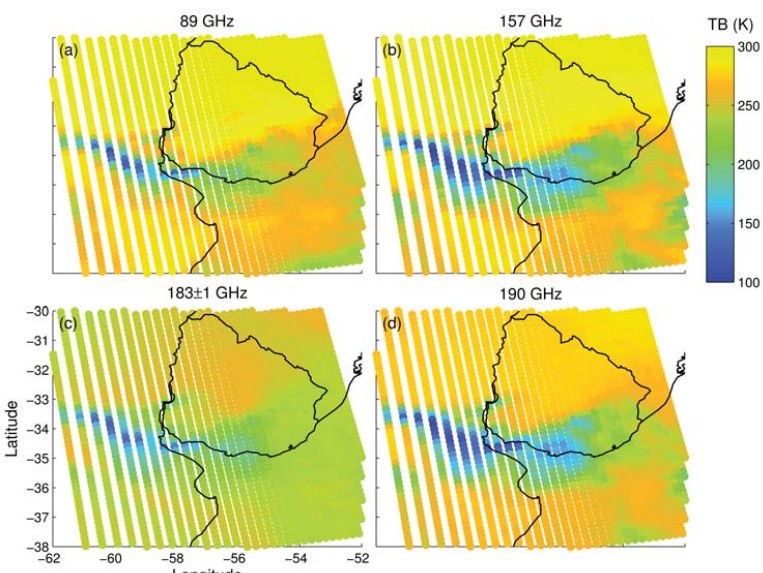

**Figure 2.**   MHS observations on 6 December 2012 at 19:00 UTC for an MCS event of interest
in the present study. Note that the 183±3 GHz channel is not shown.

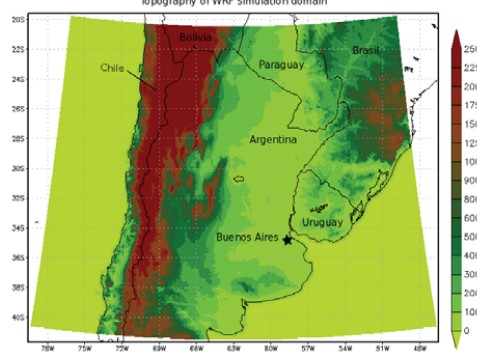

**Figure 3.**   The geographical domain used in WRF model runs illustrated by the topography of
the region in meters.



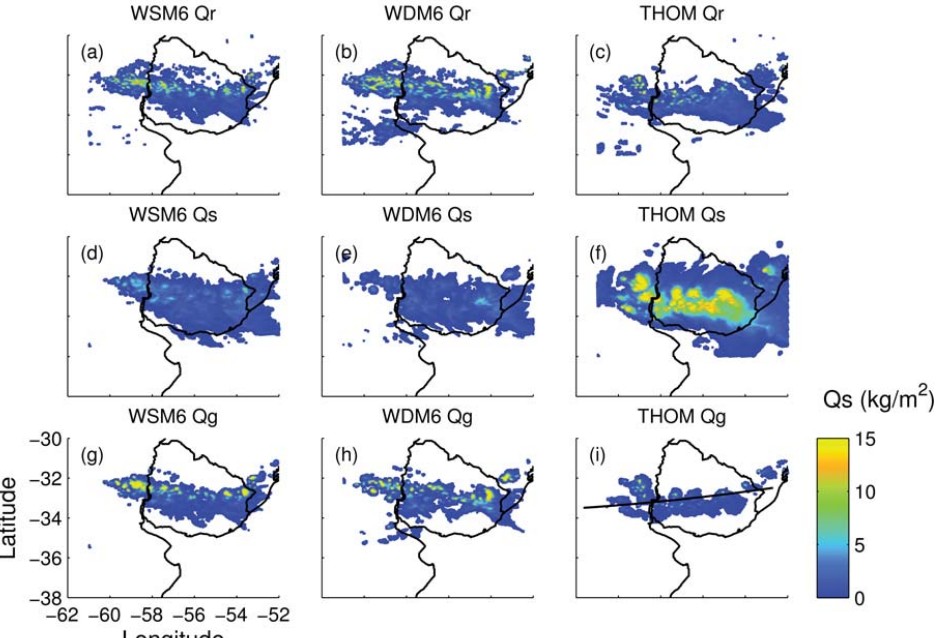

**Figure 4.** The integrated column contents in kg/m$^2$ for rain, snow and graupel, as simulated by the WRF microphysics options WSM6, WDM6 and THOM, at 1900 UTC with a 0.05 kg/m$^2$ minimum threshold. Note that cloud water and cloud ice are not shown. The black solid line in 4(i) represents an MHS transect explored in Section 4.

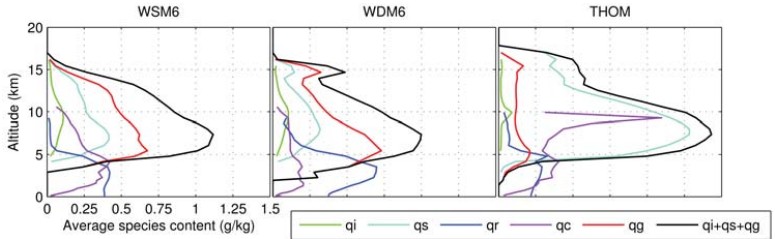

**Figure 5.** The domain-averaged vertical species content as modelled by WRF between 18:00 and 19:00 UTC by the WSM6, WDM6, and THOM microphysics options. Units are in g/kg for all species, and the domain-average is calculated from Figure 4.



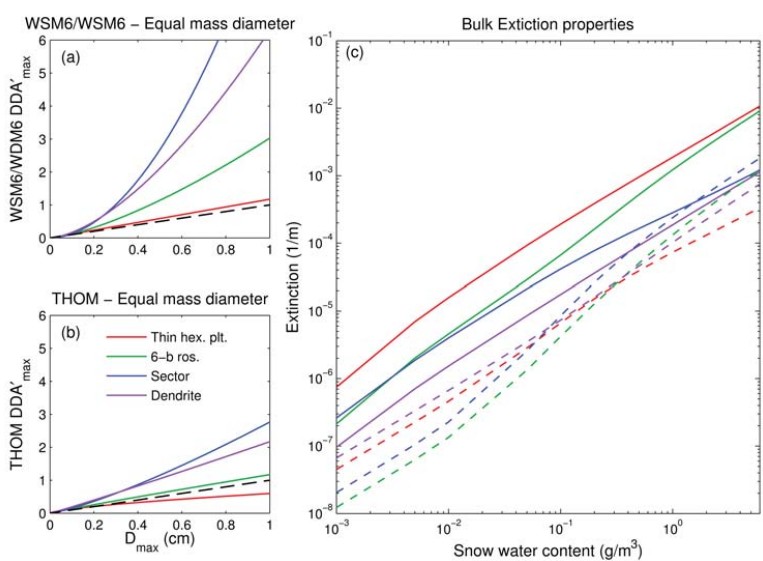

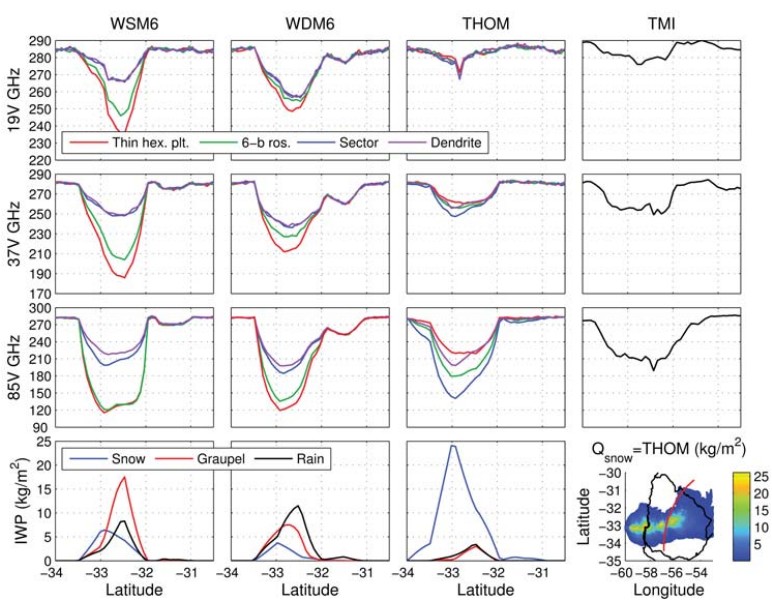

**Figure 6.** Left: The corresponding equal mass DDA habit size calculated from Equation 3 for WRF (a) WSM6 and WDM6 and (b) THOM schemes. (c) Right: The bulk scattering properties, i.e., the extinction coefficient for the WSM6 and WDM6 and the THOM schemes as a function of snow water content at 150 GHz at 263 K. The bulk optical properties have been computed by integrating the scattering properties of all equal mass *Liu* [2008] particle habits over the size distributions of interest.

**Figure 7.** The simulated brightness temperatures for the TMI 19V, 37V, 85V GHz channels along a specific transect of interest shown in the bottom right panel, using selected Liu (2008) DDA habits (see legend) and the WSM6, WDM6 and THOM WRF schemes (the first 3 columns) and the observed brightness temperatures (in black in the last column). The corresponding integrated mass contents of snow, graupel and rain are shown in the bottom row. Note that the bottom right panel shows the column integrated WRF(THOM) snow mass content for the whole scene together with a solid red line to illustrate the location of the transect of interest.

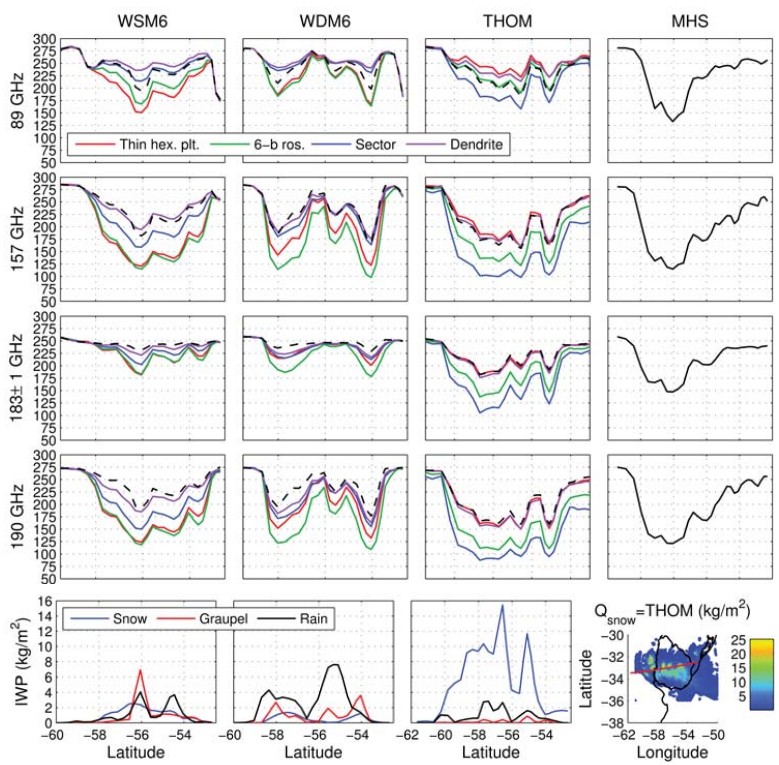

**Figure 8.** Similarly to Figure 7, the simulated brightness temperatures for the 89, 157, 183±1 and 190 GHz MHS channels along the transect of interest shown in Figure 4(i) using a selection of Liu (2008) DDA habits and the WSM6, WDM6 and THOM WRF schemes (in the first three columns). The last column shows reference MHS observations for the transect in solid black lines. The corresponding integrated mass contents of snow, graupel and rain are shown in the bottom row. Note that the bottom right panel shows the column integrated WRF(THOM) snow mass content for the whole scene together with a solid red line to illustrate the location of the transect of interest.





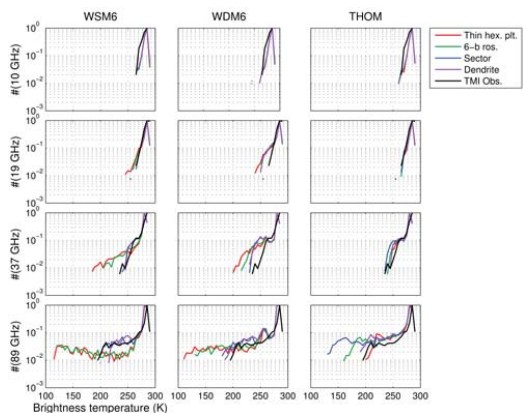

**Figure 9.** The observed (solid black line) and simulated (solid colored lines) TMI brightness temperature distributions (built with 5 K bins and where bins with less than 5 counts are neglected).

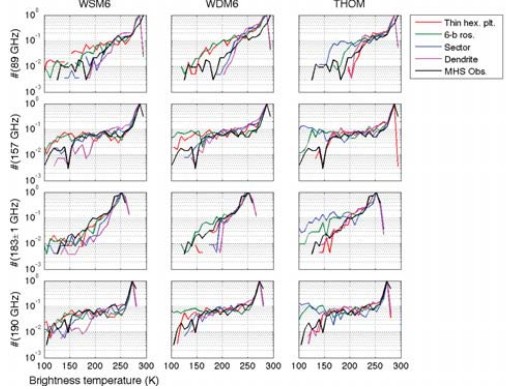

**Figure 10.** The observed (solid black line) and simulated (solid colored lines) MHS brightness temperature distributions (built with 5 K bins and where bins with less than 5 counts are neglected).





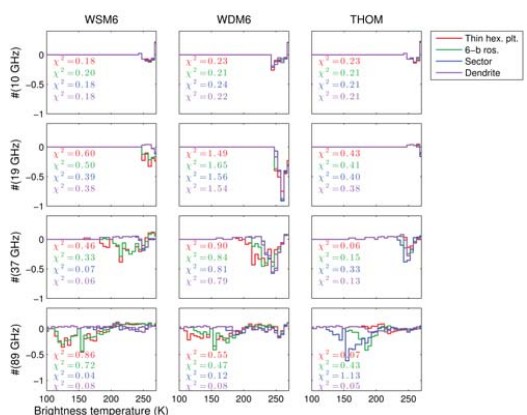

**Figure 11.** The simulated (solid colored lines) residuals of the Chi-squared test for the TMI
brightness temperature distributions. Note that the $\chi^2$ value is included for each of the DDA
habit simulated distributions calculated from all temperature bins below 270 K.





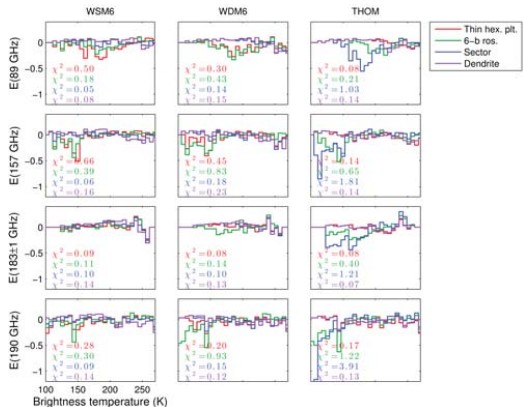

**Figure 12.** The simulated (solid colored lines) residuals of the Chi-squared test for the MHS
brightness temperature distributions. Note that the $\chi^2$ value is included for each of the DDA
habit simulated distributions calculated from all temperature bins below 270 K (250 K for the
183±1 GHz channel).



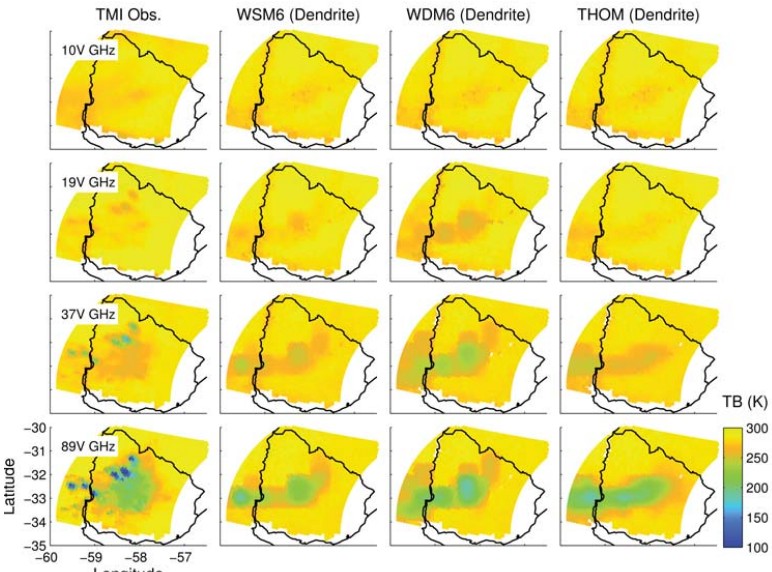

**Figure 13.**   TMI observations at 10V, 19V, 37V and 89V GHz (first column), as compared to
the corresponding radiative transfer simulations using the dendrite habits for the WSM6, WDM6
and THOM scheme simulations.





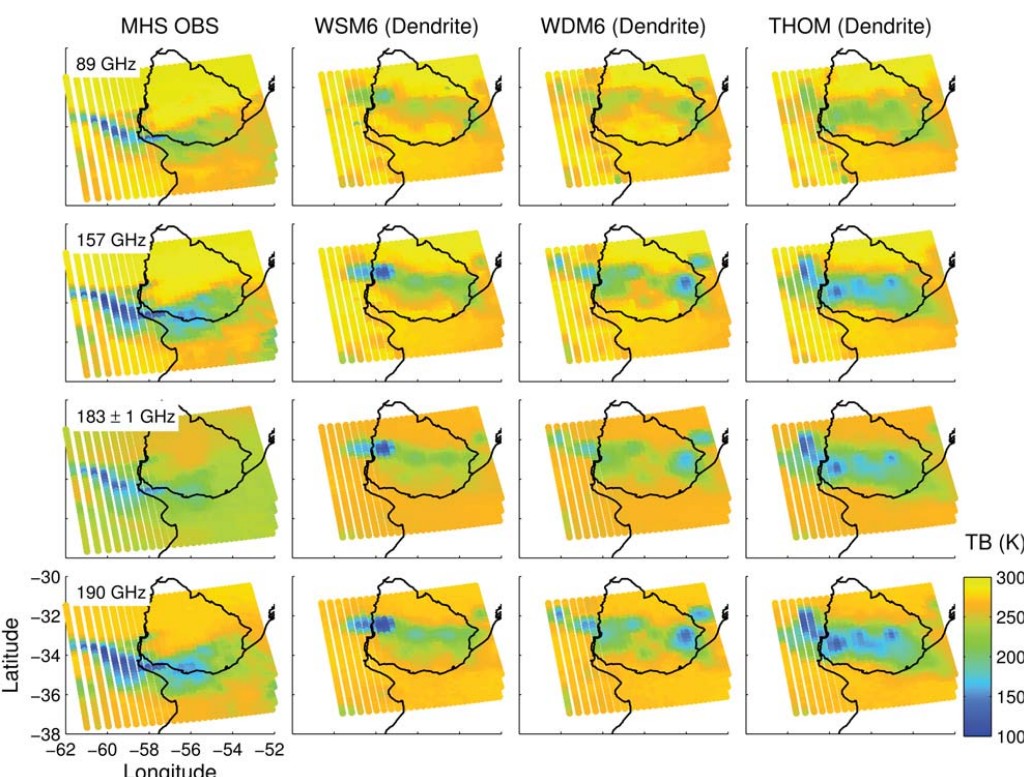

**Figure 14.** MHS observations at 89, 157, 183±1 and 190 GHz, as compared to the corresponding radiative transfer simulations using the dendrite habits for the WSM6, WDM6 and THOM scheme simulations.





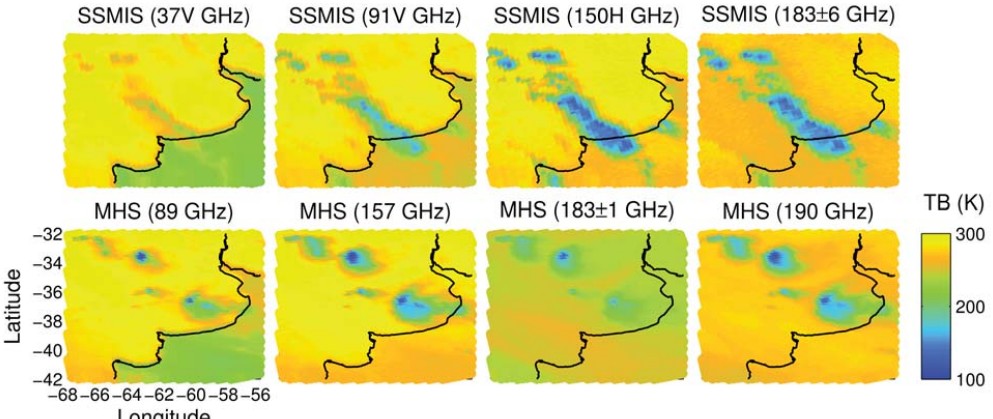

**Figure 15.** Coincident microwave observations for two MCS events of interest. Top row: observed brightness temperatures for selected SSMI/S channels over South Easter South America on the 13 January 2011 at 22 UTC. Bottom row: observed brightness temperatures for selected MHS channels over South Easter South America on the 23 January 2014 at 2 UTC.

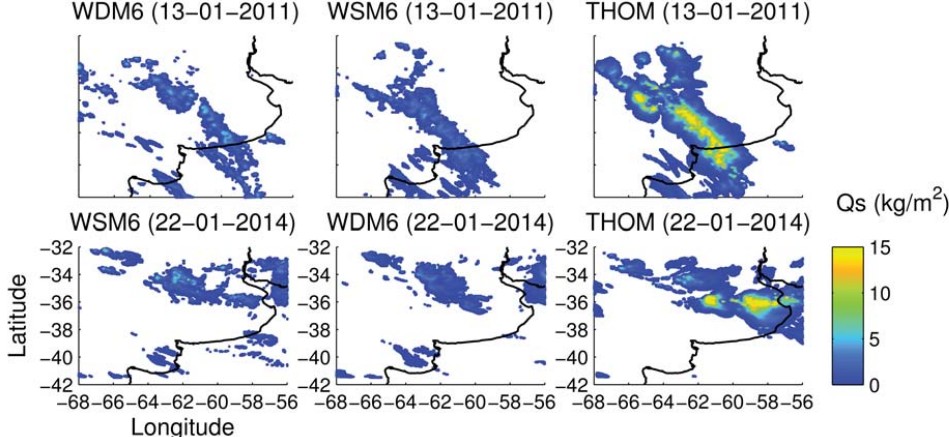

**Figure 16.** The integrated column contents in kg/m$^2$ for snow as simulated by the WRF microphysics options WSM6, WDM6 and THOM, on the 13 January 2011 at 22 UTC (top row) and on the 23 January 2014 at 2 UTC (bottom row), with a 0.05 kg/m$^2$ minimum threshold





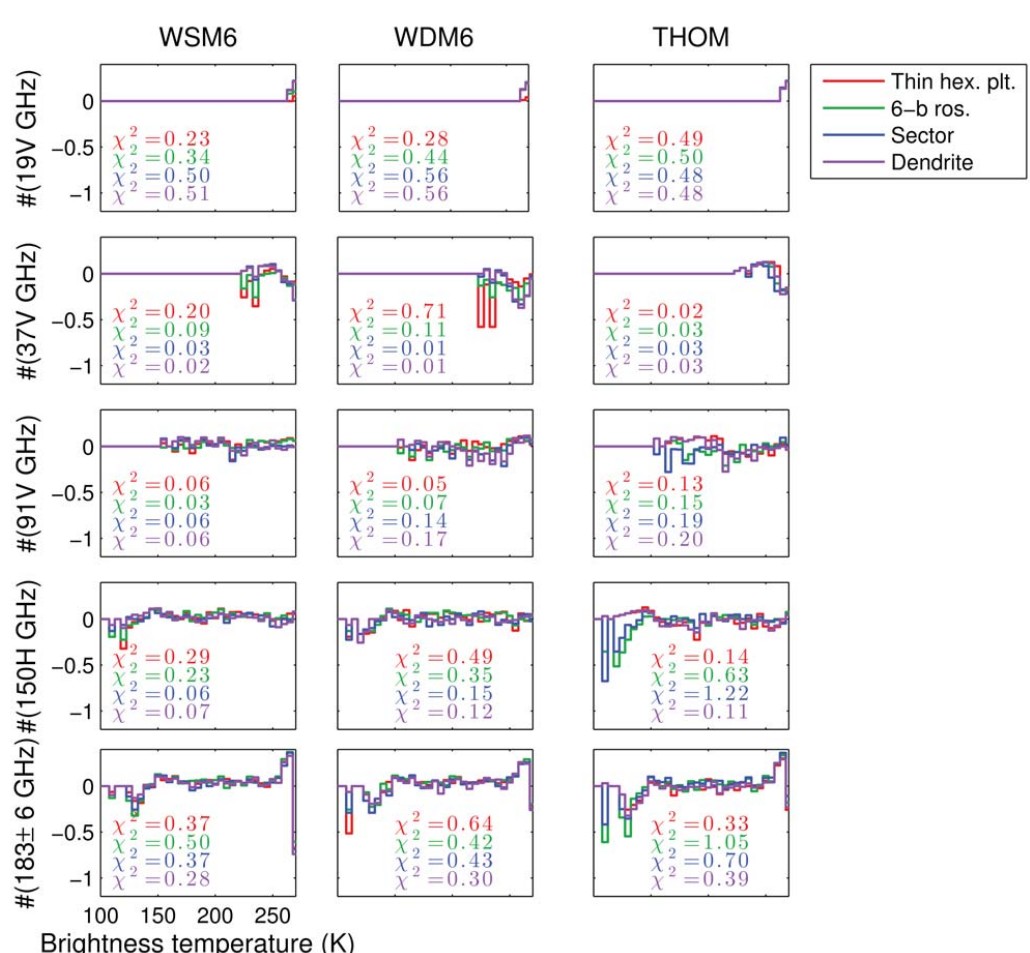

**Figure 17.** The simulated (solid colored lines) residuals of the Chi-squared test for the simulated SSMI/S 19V, 37V, 91V, 150H and 183±6 GHz channels for the MCS events on the 13 January 2011 at 22 UTC. Note that the $\chi^2$ value is included for selected DDA habit simulated distributions calculated from all temperature bins below 270 K.





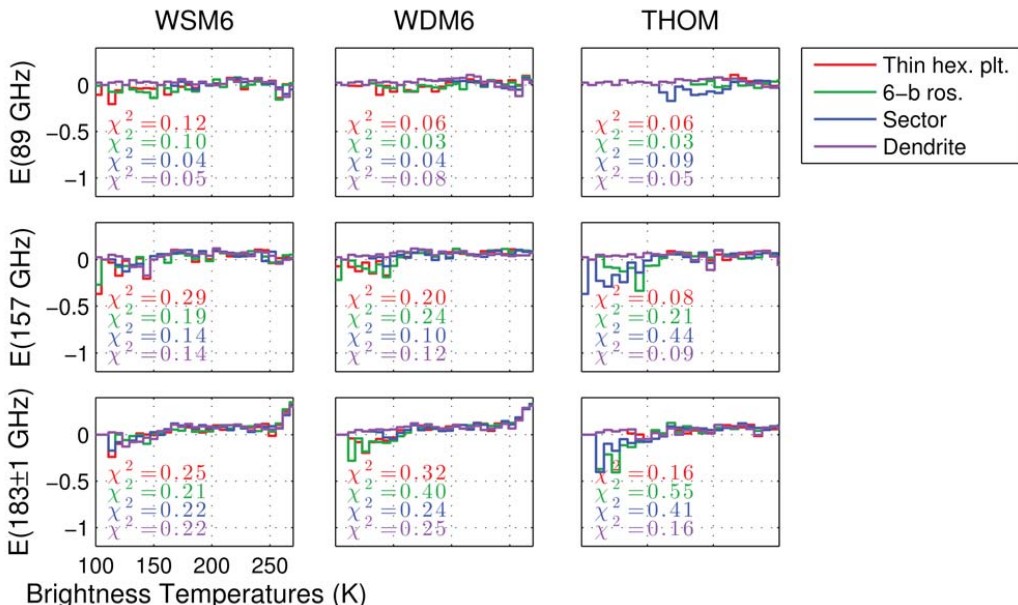

**Figure 18.** The simulated (solid colored lines) residuals of the Chi-squared test for the simulated MHS 89, 157 and 183±1 GHz channels for the MCS events on the 13 January 2011 at 22 UTC. Note that the $\chi^2$ value is included for selected DDA habit simulated distributions calculated from all temperature bins below 270 K and 250 K for the 183±1 GHz channel).



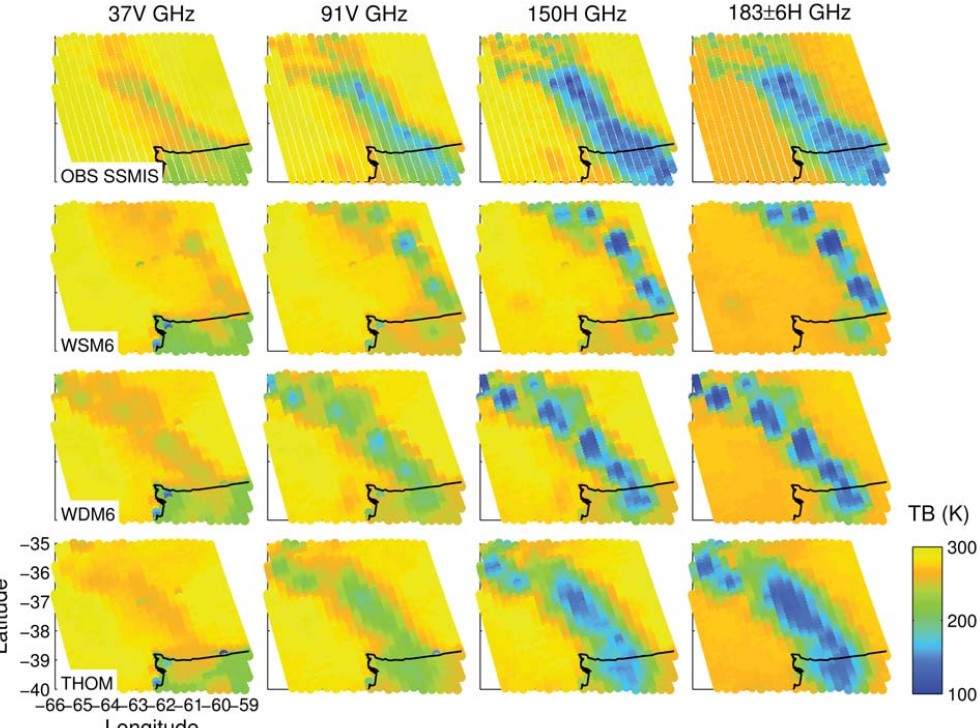

**Figure 19.** SSMI/S observations, as compared to the corresponding radiative transfer simulations using the dendrite habits for the WSM6, WDM6 and THOM scheme simulations for the 13 January 2011 event analysed.



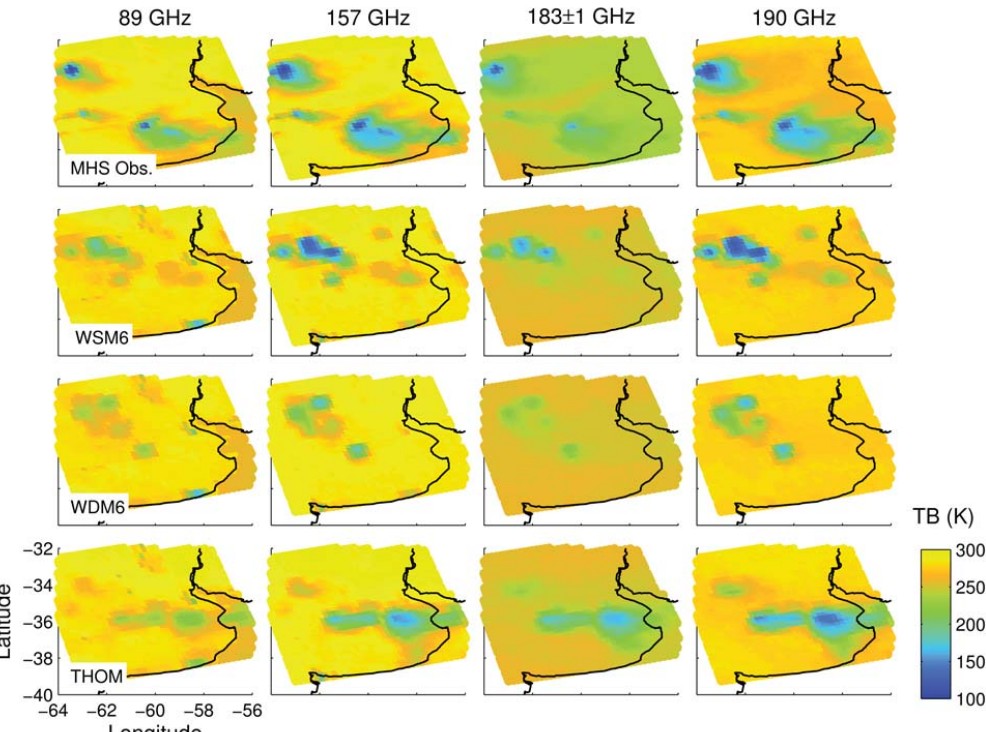

**Figure 20.**    MHS observations, as compared to the corresponding radiative transfer simula-
tions using the dendrite habits for the WSM6, WDM6 and THOM scheme simulations for the 23
January 2014 event analyzed.



798

**Table 1.** Overview of the *Liu* [2008] database

| Habit | Range of max dimension ($\mu$m) | a | b |
|---|---|---|---|
| Long hexagonal column | 121 - 4835 | 37.09 | 3.00 |
| Short hexagonal column | 83 - 3304 | 116.12 | 3.00 |
| Block hexagonal column | 66 - 2532 | 229.66 | 3.00 |
| Thick hexagonal column | 81 - 3246 | 122.66 | 3.00 |
| Thin hexagonal column | 127 - 5059 | 32.36 | 3.00 |
| 3-bullet rosette | 50 - 10000 | 0.32 | 2.37 |
| 4-bullet rosette | 50 - 10000 | 0.06 | 2.12 |
| 5-bullet rosette | 50 - 10000 | 0.07 | 2.12 |
| 6-bullet rosette | 50 - 10000 | 0.09 | 2.13 |
| Sector snowflake | 50 - 10000 | 0.002 | 1.58 |
| Dendrite snowflake | 75 - 12454 | 0.01 | 1.90 |

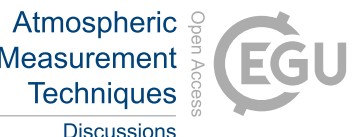

799

**Table 2.** The WRF paramterizations used

| Physics | Parametrization |
| --- | --- |
| Microphysics | WRF Single-Moment 6 (WSM6; *Hong and Lim* [2006]) |
| | WRF Double-Moment 6 (WDM6, *Hong et al.* [2010]) |
| | Thompson (THOM, *Thompson et al.* [2008]) |
| Long wave radiation | RRTM [*Mlawer et al.*, 1997] |
| Short wave radiation | Dudhia [*Dudhia*, 1989] |
| Surface-layer exchange coefficient | Monin-Obukhov (Janjic Eta) scheme |
| Surface processes | Noah LSM [*Chen and Dudhia*, 2001] |
| PBL | MYJ Janjic [*Janjic*, 1994] |





## Acknowledgments

This research was funded by the CNRS LEFE/IMAGO program. The authors would like to express their gratitude to the ARTS community for developing and maintaining an open source distribution

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

    the most intense thunderstorms on earth?, *Bulletin of the American Meteorological
    Society*, *87*(8), 1057.