# Peer review of "Analysis and evaluation of WRF microphysical schemes for deep moist convection over Southeastern South America (SESA) using microwave satellite observations and radiative transfer simulations"

_Atmospheric Measurement Techniques, 2017_

## Referee Comment (RC1) · A.J. Geer (Referee) · 31 May 2017

This is an interesting study evaluating the ability of WRF simulations to produce realistic convection simulations, using the model-to-satellite approach, with satellite-observed microwave radiances as the reference. A first highlight of the paper is the novel 'equal mass approach' for converting WRF model hydrometeor particles to their equivalents in the forward radiative transfer. This is a pragmatic and useful way to match up particles that may have different geometric representations in the moist physics and the radiative transfer. A second highlight is the evaluation of the WRF microphysics schemes, indicating substantial differences between WSM6/WDM6 and Thompson microphysics schemes. The manuscript is generally fit for publication, but there a few main points for the authors to consider:

1) In the introduction, or at the end of section 2.2, it would be good to survey all previous validation of the WRF microphysics and to summarise any known issues.

2) A minor but repeated issue (e.g. lines 142-144; lines 468, 478, 495) is the attribution of brightness temperature (TB) depressions at low frequencies (e.g. 85 GHz or less) uniquely to scattering. Over land, at low frequencies, cloud water and particularly rain (and possibly snow too) can generate TB depressions through absorption and emission pushing the weighting function up to colder layers in the atmosphere. It would be good to examine more closely whether it really is scattering causing the TB depressions in all cases. Incorrect modelling of the cloud water or rain could also contribute to mismatches between observations and simulations.

3) A feature of the equal-mass approach is that it changes the relative amount of scattering generated by the Liu particles, in one case making the sector snowflake the most scattering particle (e.g. lines 506-510; 537-541). This is one of the most interesting aspects of the study and it could do with further exploration in the text (and possibly more figures) to explain exactly how this occurs (note the lack of labelling on Fig. 6 has not helped here - minor point 8).

4) Some of figures 16-20 could be considered for reduction, as they mainly repeat and confirm the results of the case study in section 4.

5) Some figures are too small and fuzzy - e.g. Figure 3 longitude and latitude legends; Figs 9, 10, 11, 12.

Minor issues

1) Line 232: "As expected WSM6 and WDM6 schemes model similar ...loadings": for

the benefit of the reader, could the reason be restated here, instead of just saying "as expected"?

2) Line 308-311: "The only computationally realistic approach is to assume a one-shape model". This statement could be challenged: an ensemble of particles could be used without much additional computational effort - for example something like the Baran (2009) ensemble.

3) Lines 338-339, 445-447. In both these areas the question arises "are both snow and graupel simulated using the same Liu particle habit?" The answer is probably yes, but it would be worth (re?)-stating this for the benefit of the reader.

4) Line 524: "The higher the window channel" - higher what? frequency?

5) Line 602-603: "WDM6 leads to excesive scattering at > 19 GHz". This is not obvious to me. At 37 GHz WDM6 is the only model to generate TB depressions as low as observed, albeit over a wider area than observed. At 89 GHz, none of the schemes generates sufficient TB depression.

6) Line 603-604: "Figure 14 shows good agreement" - this could be restated in more depth and a little more critically. For example there is the broader spread of TB depressions generated by THOM, versus perhaps too-narrow areas of TB depression from the other schemes. As in Fig. 13, none of the schemes have deep enough depressions at 89 GHz.

7) Line 709-710: poor wording in this sentence suggests that the Liu habits all have the same bulk scattering properties - please rephrase.

8) Figure 6 has no key for the black dashed line in the left panels, or the significance of solid versus dashed in the right panel.

9) Figure 8 needs a key to the black dashed line.

10) The many figures featuring chi-squared ststs are not all consitent in terms of the y

axis labelling: some use "#()" (not explained) and some "E()" (explained in the text)

Grammar points

1) There is a repeated error in the text that writes "South Easter" instead of the correct "South Eastern"

2) Line 165: "described" should be "describe"

3) In a few places, "sensibility" is used in place of the correct english term "sensitivity".

4) Line 230 "similarly to TMI observations" is hard to understand and the sentence should be rewritten for clarity.

References

Baran, A.J., Connolly, P.J. and Lee, C., 2009. Testing an ensemble model of cirrus ice crystals using midlatitude in situ estimates of ice water content, volume extinction coefficient and the total solar optical depth. Journal of Quantitative Spectroscopy and Radiative Transfer, 110(14), pp.1579-1598.

---

## Referee Comment (RC2) · Anonymous Referee #2 · 16 Jun 2017

General comments: This study provides some interest analysis on the scattering properties of frozen particles in the WRF model simulations. The WRF simulations are converted to microwave brightness temperature with an equal mass habit approach, and compared with satellite observations. Three microphysics parameterizations and their sensitivity to different single scattering properties of frozen particles are investigated. I would recommend the manuscript for publication after the following comments are addressed.

[Figure]

Major comments: 1. Line 181-190: Is there any specific reason to select these three microphysics schemes? Or just randomly? Why are both the WSM6 and WDM6 schemes selected? This study targets on frozen particles. However, WSM6 and WDM6 use the same parameterization of frozen particles. The performance of WSM6 and WDM6 are consistent for most of the results shown in the manuscript. Is it necessary to include both of them?

2. Figure 7 and 8: As mentioned in Line 414-415, there are differences in the location of the observed and modelled cloud system. Is it representative to discuss the differences among simulations and observations? For example, the difference of IWP (graupel) between WSM6 and WDM6 are large for the transect in Fig. 7. However, the difference of graupel is small between WSM6 and WDM6 in Fig. 5. It will be more representative to use zonal/meridional means for comparison. And it will be interesting to see the relative contribution (sensitivity) of snow/graupel to the simulated brightness temperature in different microphysics schemes.

3. As one of the goals of this study is to evaluate the microphysics parameterizations, could the authors have more discussions about how to interpret/use these results in terms of evaluation? As shown in the manuscript, there are large uncertainties in distribution, mass, and scattering properties of frozen particles in different microphysics schemes. However, all the simulations produce comparable bright temperature to the observations. Can we conclude from this study which scheme produces more realistic frozen particles?

Minor comments: 1. Line 173 "the five hydrometeor categories": It depends on the selected microphysics scheme, for example, WSM3 does not provide five hydrometeor categories.

2. Line 204-207: It is not easy to follow. It will be helpful for reader to understand by providing the following information shown in Thompson et al. (2008): "the spherical and constant-density snow assumption is applied in models through the assumed mass-

diameter relation, usually with the power law." "The new scheme considers snow to be primarily composed of fractal-like aggregated crystals, which likely captures the vast majority of the actual snow mass reaching the earth's surface."

3. Line 237-238: THOM has more frozen particles than WSM6 and WDM6.

4. Line 244-246: Is there any reference?

5. Figure 6C: Please add legend.

---

## Author Comment (AC1) · 1 Jul 2017

We would like to thank Alan Geer for his insightful comments. Below we address each of the comments made.

Main points

1) In the introduction, or at the end of section 2.2, it would be good to survey all previous

validation of the WRF microphysics and to summarise any known issues.

Section 2.2 has been modified to include more information on previous validations and how these compare with simulations in the present work. A more in-depth review of all previous validation studies is outside the scope of this paper. Please see Section 2.2 in the new paper version. I copy below the new paragraph:

"Validation techniques of these schemes depend upon the availability of observations. In-situ measurements are essential for detailed and direct microphysics validations, such as particle size distributions and liquid/ice water content. However, these observations are limited to certain field campaigns as well as certain parts of the storms. On the other hand, satellite observations can cover this gap if they are widely available and are very useful for model validation. Figure 4 shows that the WSM6 and the WDM6 schemes model similar hydrometeor mass loadings and storm morphology. This is expected as the WDM6 is developed from the processes in the WSM6 scheme. The THOM scheme, on the other hand, shows much higher snow contents as reported in many studies (e.g., Kim et al., 2013 and Gallus and Pfeifer, 2008). Figure 5 further shows the domain-averaged vertical distribution of the hydrometeor contents modelled by the different schemes between 18:00 and 19:00 UTC. Units are in g/kg for all the species. Both Figure 4 and Figure 4 show a comparable behaviour in the frozen phase (ice, snow and graupel) in the WSM6 and WDM6 schemes. This is expected because the WDM6 scheme follows the cold-rain processes of the WSM6 scheme and the added processes in the WDM6 do not affect the frozen phases directly (Lim et al, 2010). Comparing the warm-rain processes of the WSM6 and WDM6 schemes, Figure 5 shows an increase of the WDM6 rainwater mixing ratio below 5 km with less cloud droplet mixing ratios, as reported by Kim et al., 2013 who studied a typhoon event and reported that the liquid phase in the WDM6 scheme produced a significantly larger amount of rainwater but smaller cloud droplet mixing ratio. Various studies have shown that the double-moment approach in the WDM6 scheme may help to achieve a more realistic simulation of convective rain and rainfall retrievals, as the rain number concentration plays an important role in determining the precipitation rate and storm morphology because it modulates the related microphysics terms, in particular, the evaporation rate (Morrison et al., 2009, Li et al., 2009a,b, Lim and Hong, 2010). Figure 4 shows that the the THOM scheme predicts the smallest amount of rain water, while Figure 5 further shows that the THOM scheme is dominated by snow throughout the vertical profile. These conclusions are also reached by Kim et al., 2013. The THOM scheme has a maximum cloud water content between 8 and 10 km. This peak of enhanced cloud water content is found within and around strong convective updrafts (Otkin et al., 2003). In order to compare the distribution of the frozen hydrometeor species among the total frozen phase for each scheme, Figure 5 additionally shows the mean vertical profile of the total frozen content (i.e., ice+snow+graupel, shown in light blue). The total frozen content is comparable in magnitude in all the schemes analyzed but since each scheme has different intrinsic assumed characteristics and microphysical processes, they partition the total content in different ways between graupel, cloud ice, and snow. The THOM scheme has the most prominent vertical structure. Note that very similar remarks can be drawn from the model simulations at 07:00 UTC in coincidence with the available TMI observations (not shown)."

2) A minor but repeated issue (e.g. lines 142-144; lines 468, 478, 495) is the attribution of brightness temperature (TB) depressions at low frequencies (e.g. 85 GHz or less) uniquely to scattering. Over land, at low frequencies, cloud water and particularly rain (and possibly snow too) can generate TB depressions through absorption and emission pushing the weighting function up to colder layers in the atmosphere. It would be good to examine more closely whether it really is scattering causing the TB depressions in all cases. Incorrect modelling of the cloud water or rain could also contribute to mismatches between observations and simulations.

Thank you for raising this point up for discussion. I agree that at low frequencies, cloud water and rainwater can generate TB depressions through absorption and emission by pushing the weighting functions up to colder layers in the atmosphere. In fact, there is

a contribution of these processes in the cases examined, especially at the lowest frequencies (especially below 37 GHz). This was examined more closely, but not shown. The transects simulated in Figure 7 for TMI were simulated with no population of frozen species. For example, for WSM6(DDA=sector) simulations at the 19 GHz channel, TBs simulated without the frozen phase did not change significantly (5K warmer TBs). This means that at 19 GHz the most important contribution to the TB depressions simulated is liquid water absorption, emissions and scattering. These liquid phase only TBs simulated with the WSM6 scheme are 10K colder than those observed by TMI. This bias is larger for the WDM6 scheme which has a larger simulated rain water path. This is the order of magnitude that incorrect modelling of rain water can contribute to biases in simulated TBs at 19 GHz. The same test was run for the THOM scheme. For the THOM scheme, TBs simulated without the frozen phase changed even less significantly (approx. 3K warmer TBs for DDA=sector habits) and remained closed to the observed TBs. At 37 GHz, a 10K difference is observed between simulations with and without the frozen phase for WSM6(DDA=sector), and both simulations are comparable to the observed TBs as the observed TBs lie between the simulations with and without the frozen phase for the WSM6(DDA=sector). For the THOM scheme, TBs simulations without the frozen phase were 12K warmer than those with the frozen phase. Considering that the THOM scheme simulations with all the phases are shown to be close to the observed TMI observations, those simulations without the frozen phase were shown to be warmer than those observed. At 85 GHz, however, the contribution of frozen scattering is much stronger and a 60K difference is found between simulations with and without the frozen phase for WSM6(DDA=sector). TB simulations without the frozen phase are 60 K warmer than those observed by TMI. This shows that frozen phase scattering is key to explain the observed TBs. The DDA sector habit has been shown to be the least scattering habits when used through the equal mass approach with the WSM6 scheme. If the above exercise is repeated with DDA=6-b rosettes for example, differences between the simulations with and without the frozen phase are much larger. At 85 GHz, it is evident that scattering causes the TB depressions. A similar behaviour

is observed for the THOM scheme simulations. The higher the microwave frequency of the (window) channel, the higher the sensitivity to scattering of the frozen phase, but even a the 37 GHz channel, the 10K difference is an important contribution from frozen scattering. As discussed above, it can be debated that the WSM6/WDM6 schemes show incorrect modelling of the liquid water phase from the 19GHz simulations. However, the uncertainties associated with incorrect modelling of the cloud water and rain are much smaller than those associated with incorrect modelling of the frozen phase as evidenced by the large sensitivity to snow shape for example. In addition, this are extremely severe convection cases with a large IWP. At 19 GHz, differences are due to the incorrect modelling of the liquid water phase in the WSM6/WDM6.

To be more thorough in the text the following modifications have been made:

Lines 142-144: The text reads "The highly scattering MCS event is evidenced by brightness temperature depressions at the higher frequency channels ($\geq$37 GHz). At the lower frequency channels ($\leq$37 GHz), TMI is mostly sensitive to surface emission, and cloud absorption and emission."

Lines 468, 478: These lines, deals with the differences in the WSM6/WDM6 simulations with different DDA habits. These differences in TB arise from the scattering signal of graupel+snow, which are strong enough to be seen at 19 GHz. Simulations without the frozen phase show that simulations are up to 10K colder for the sector snowflakes. This means that a bias also exists due to the liquid phase WRF model outputs. This is not observed in the THOM scheme, where simulations at 19 GHz with and without the frozen phase are very close to those observed by TMI. The text has been slightly modified: "At 19 GHz, all DDA habits produce excessive scattering for the WSM6 and WDM6 simulations, where the dendrite and sector habits simulate the warmest TBs closest to the observed reference TBs, and the thick hexagonal plates and the block, long and short hexagonal columns (not shown) are the most scattering habits, producing the coldest TBs, followed by the thin hexagonal plate and the rosettes (only the 6-b rosette is shown). On the other hand, all DDA habits in the THOM scheme simulations produce

similar TB depressions to those observed. The large depression observed at 19 GHz in the WSM6/WDM6 simulations is due to the high IWP graupel frozen phase contents simulated by WRF. Simulations for the WSM6 show larger brightness depressions at 19 GHz as they have a larger IWP graupel content. Note that simulations without the frozen phase show simulated brightness temperatures closer to those observed, but still show a significant cold bias (10 K). Note that due to the small brightness temperature depressions simulated using the THOM scheme, the signal coming from the lake at approximately -32.9$ˆo$ can be observed at 19 GHz, while simulations using the WSM6/WDM6 schemes are dominated by excessive scattering and consequently frozen phase scattering cloud signals dominate all surface signals. Note that although the THOM scheme is predicting the largest amount of integrated snow content, it does not necessarily produce the largest brightness temperature depressions."

3) A feature of the equal-mass approach is that it changes the relative amount of scattering generated by the Liu particles, in one case making the sector snowflake the most scattering particle (e.g. lines 506-510; 537-541). This is one of the most interesting aspects of the study and it could do with further exploration in the text (and possibly more figures) to explain exactly how this occurs (note the lack of labelling on Fig. 6 has not helped here - minor point 8).

Thank you for bringing up this central issue for discussion. Figure 6 has been modified and a futher figure has been added (Both attached in this response). Please below the legends for these figures:

[revised manuscript text omitted]

4) Some of figures 16-20 could be considered for reduction, as they mainly repeat and confirm the results of the case study in section 4. I think that they are useful for reference of how the different cases behave in terms of the radiative transfer simulations.

5) Some figures are too small and fuzzy - e.g. Figure 3 longitude and latitude legends; Figs 9, 10, 11, 12. Some figures have been made larger. The fuzzy effect I believe is

due to the quality of figures downgraded at some point.

Minor points

1) Line 232: "As expected WSM6 and WDM6 schemes model similar ...loadings": for the benefit of the reader, could the reason be restated here, instead of just saying "as expected"?

This paragraph has been modified based on RC1 Point (1) and the text now reads: "Both Figure 4 and Figure 5 show a comparable behaviour in the frozen phase (ice, snow and graupel) in the WSM6 and WDM6 schemes. This is expected because the WDM6 scheme follows the cold-rain processes of the WSM6 scheme and the added processes in the WDM6 do not affect the frozen phases directly (Lim and Hong, 2010)"

2) Line 308-311: "The only computationally realistic approach is to assume a one shape model". This statement could be challenged: an ensemble of particles could be used without much additional computational effort - for example something like the Baran (2009) ensemble. http://ac.els-cdn.com/S0022407309000661/1-s2.0-S0022407309000661-main.pdf?_tid=2084b234-506d-11e7-b2db-00000aacb360&acdnat=1497381708_72b9189f7ca88301717bad59c21a3295

The model in Baran (2009) consists of six ice crystal shapes for cirrus clouds: a simple hexagonal ice column and a six-branched bullet-rosette (the smallest ice crystals in the PSD). As the ice crystal maximum dimension increases so does the ice crystal complexity by forming aggregates of hexagonal ice columns, which are arbitrarily attached to other hexagonal elements, forming three, five and eight element aggregates until finally, a chain of 10 hexagonal elements is constructed. This 10 element hexagonal ice aggregate represents the largest ice crystals in the PSD. The PSD is sub-divided into six equal sections with each ice crystal shape distributed within each section, so the simple hexagonal ice column and 10 element ice aggregate is distributed within the first and sixth sections of the PSD. It is true that something like this could be implemented in the context of larger hydrometeors, not pristine ice crystals like in cirrus clouds, without

additional computational effort. This is however, outside the scope of the paper. The present work exploited the existent Liu (2004) DDA scattering databases, and choosing different shapes for different size ranges would be arbitrary and lack physical support. We strongly agree that an ensemble could be developed and tested without additional computational effort.

3) Lines 338-339, 445-447. In both these areas the question arises "are both snow and graupel simulated using the same Liu particle habit?"The answer is probably yes, but it would be worth (re?)-stating this for the benefit of the reader.

Yes. This has been re-stated again.

4) Line 524: "The higher the window channel" - higher what? Frequency?

Thank you! This has been corrected to "The higher the frequency of the window channel"

5) Line 602-603: "WDM6 leads to excessive scattering at > 19 GHz". This is not obvious to me. At 37 GHz WDM6 is the only model to generate TB depressions as low as observed, albeit over a wider area than observed. At 89 GHz, none of the schemes generates sufficient TB depression.

This is true. It has been removed.

6) Line 603-604: "Figure 14 shows good agreement" - this could be restated in more depth and a little more critically. For example there is the broader spread of TB depressions generated by THOM, versus perhaps too-narrow areas of TB depression from the other schemes. As in Fig. 13, none of the schemes have deep enough depressions at 89 GHz.

The following has replaced those lines: "Despite errors in the location and coverage of the spatial structures of the cloudy fields modelled by WRF, the results depicted show that the three WRF microphysics schemes can be used to simulate the observed brightness temperature depressions provided special care is taken to represent the

scattering properties of the snow and graupel species. At 19 GHz, the THOM scheme does not have deep enough brightness temperature depressions as observed, while at 37 GHz, the WDM6 scheme is the only model to generate brightness temperature depressions as low as observed, albeit over a wider area than observed. At 89 GHz, none of the schemes reach deep enough brightness temperature depressions as observed. MHS simulations have a higher sensitivity to frozen scattering. Figure 15 shows good agreement between the three microphysics schemes and MHS observations. The THOM scheme, however, has a broader spread of TB depressions, versus the too-narrow areas of TB depression from the other schemes. Similarly to Figure 14, Figure 15 also shows that at 89 GHz none of the schemes reach deep enough brightness temperature depressions as observed."

7) Line 709-710: poor wording in this sentence suggests that the Liu habits all have the same bulk scattering properties - please rephrase.

The sentence "The bulk scattering properties of the Liu (2008) habits are similar for the WSM6 and WDM6 schemes, but different to the THOM scheme" has been rewritten as "The resultant bulk scattering properties of each of the Liu (2008) habits under the WSM6 scheme is similar under the WDM6 scheme, but different under the THOM scheme."

8) Figure 6 has no key for the black dashed line in the left panels, or the significance of solid versus dashed in the right panel.

Thank you. This has also been raised by the other reviewer. Thi Figure has been modified in reference to RC1 point (3).

9) Figure 8 needs a key to the black dashed line.

Thank you. This has been added. It is also referenced in the legend.

10) The many figures featuring chi-squared tests are not all consistent in terms of the y axis labelling: some use "#()" (not explained) and some "E()" (explained in the text)

Thank you for pointing this out. This has been corrected to use Ei as used in the text. The "#()" only belonged to the histograms.

Grammar points

1) There is a repeated error in the text that writes "South Easter" instead of the correct "South Eastern"

Thank you for pointing this out. It has been corrected.

2) Line 165: "described" should be "describe"

Thank you for pointing this out. It has been corrected.

3) In a few places, "sensibility" is used in place of the correct english term "sensitivity".

Thank you for pointing this out. It has been corrected.

4) Line 230 "similarly to TMI observations" is hard to understand and the sentence should be rewritten for clarity.

This sentence has been rewritten as: "A close examination of MHS (and TMI) observations in Figure 2 (Figure 1) and the WRF cloud outputs in Figure 4 (not shown for TMI passage time), however, reveals that the cloud system modelled by WRF is slightly time lagged and misplaced with respect to the observations."

**Simulated WRF Snow Profile**

(a)

Height (km) vs $q_{snow}$ (g/m³)

WSM6
THOM

**Corresponding Bulk Scattering Properties at 150 GHz**

(b)

Extinction (1/m)

Thin hex. plt.
6–b ros.
Sector
Dendrite

**Fig. 1.** Figure 7

[Figure]

[Figure]

**Fig. 2.** Figure 6

---

## Author Comment (AC2) · 1 Jul 2017

We would like to thank the reviewer for his/her comments. Below we address each of the comments made.

Major comments:

1. Line 181-190: Is there any specific reason to select these three microphysics

schemes? Or just randomly? Why are both the WSM6 and WDM6 schemes selected? This study targets on frozen particles. However, WSM6 and WDM6 use the same parameterization of frozen particles. The performance of WSM6 and WDM6 are consistent for most of the results shown in the manuscript. Is it necessary to include both of them?

The WSM6 was selected because it is being used locally for various meteorological applications and for assimilation studies. The WDM6 was selected to evaluate how different simulations were for the double-moment version of the scheme. Additionally, the THOM scheme was selected given its more realistic PSD and snow density parameters. Despite the fact that the cold-rain processes in the WSM6 and WDM6 schemes are the same, the impact of the double moment scheme in the rain number concentration is large and past studies have indicated (Morrison et al., 2009, Li et al., 2009a,b, Lim and Hong 2010) that the rain number concentration plays an important role in determining the precipitation rate and storm morphology because it modules the related microphysics terms, in particular, the evaporation rate. The WDM6 scheme has been shown to improve skill statistics in precipitation forecasts (e.g., Hong and Lim 2009). Recurring evaluation of these schemes is still necessary and for this reason we consider interesting to show the results for both.

2. Figure 7 and 8: As mentioned in Line 414-415, there are differences in the location of the observed and modelled cloud system. Is it representative to discuss the differences among simulations and observations? For example, the difference of IWP (graupel) between WSM6 and WDM6 are large for the transect in Fig. 7. However, the difference of graupel is small between WSM6 and WDM6 in Fig. 5. It will be more representative to use zonal/meridional means for comparison. And it will be interesting to see the relative contribution (sensitivity) of snow/graupel to the simulated brightness temperature in different microphysics schemes.

We thank the reviewer for these comments. The main point of Figures 7 and 8 is to show the sensitivity of the transect to the different Liu (2008) DDA habits analyzed. The

real observations are shown for a reference, as the main analysis of representativity is made from Figures 9-14 with the histograms and the Chi-square test, with an analysis of the distribution of observed and simulated brightness temperatures. With regards to the last comment of the relative contribution of the frozen phase in the simulations, this has been added in Lines 513-516 and Lines 551-556 (shown in red in new manuscript version).

3. As one of the goals of this study is to evaluate the microphysics parameterizations, could the authors have more discussions about how to interpret/use these results in terms of evaluation? As shown in the manuscript, there are large uncertainties in distribution, mass, and scattering properties of frozen particles in different microphysics schemes. However, all the simulations produce comparable bright temperature to the observations. Can we conclude from this study which scheme produces more realistic frozen particles?

Thank you. The THOM scheme parameterizations in terms of snow density are more realistic than constant density WSM6/WDM6 constant density parameterizations. This discussion in the focus of ongoing work that includes ground and satellite radars. Which scheme is producing the most realistic frozen particles is an interesting question which is being addressed.

Minor comments: 1. Line 173 "the five hydrometeor categories": It depends on the selected microphysics scheme, for example, WSM3 does not provide five hydrometeor categories.

This is true. It has been corrected to: ÂÍIt provides a full description of atmospheric parameters (i.e., pressure, temperature, and prognostic water substance variables).ÂÍ

2. Line 204-207: It is not easy to follow. It will be helpful for reader to understand by providing the following information shown in Thompson et al. (2008): "the spherical and constant-density snow assumption is applied in models through the assumed mass-diameter relation, usually with the power law." "The new scheme considers snow to be

primarily composed of fractal-like aggregated crystals, which likely captures the vast majority of the actual snow mass reaching the earth's surface."

Thank you for the suggestion. The text has been modified to make this more clear: "The WSM6 and WDM6 schemes, like most models, use a spherical and constant-density snow assumption through the application of a mass-diameter relation, usually with a power law $m(D)=(ðÌIJŃ/6)sD3$, where s is the assumed fixed density of snow (for WSM6/WDM6 s=0.1kg/m3) and D is the particle diameter. Unlike most schemes, snow density in the THOM scheme is not fixed, but varies with size through the mass-size relation $m(D)=0.069D2$. This is an important difference since observational studies rarely support fixed density snow habits. Magono [1965] and many later studies recognize that a size-independent density is not a physically sound assumption for snowflakes because of the rigidity of ice and the nature of the snow formation processes (Leinonen et al.[2012]). In this sense, the THOM scheme considers snow to be primarily composed of fractal-like aggregated crystals (Thompson et al. [2008]), rather than spherical constant snow crystals, which is a much more realistic approach than the WSM6/WDM6 schemes."

3. Line 237-238: THOM has more frozen particles than WSM6 and WDM6.

We consider that the domain average vertical content is comparable. Yes, the THOM scheme has more frozen particles.

4. Line 244-246: Is there any reference?

Yes. It has been added: Otkin et al., 2003. A comparison of microphysical schemes in the wrf model during a severe weather event.

5. Figure 6C: Please add legend

This figure has been modified, and the legend carefully updated.